# Hetero-site nucleation for growing twisted bilayer graphene with a wide range of twist angles

Luzhao Sun [1,2,3,10], Zihao Wang[4,10], Yuechen Wang[1,2,10], Liang Zhao [5], Yanglizhi Li[1,2,3], Buhang Chen[3], Shenghong Huang[6✉], Shishu Zhang[1], Wendong Wang[4], Ding Pei[7], Hongwei Fang[8], Shan Zhong[1], Haiyang Liu[1], Jincan Zhang[1,3], Lianming Tong [1], Yulin Chen[7,8], Zhenyu Li [9], Mark H. Rümmeli[5], Kostya S. Novoselov [4], Hailin Peng [1,3✉], Li Lin [4✉] & Zhongfan Liu [1,3✉]

Twisted bilayer graphene (tBLG) has recently attracted growing interest due to its unique twist-angle-dependent electronic properties. The preparation of high-quality large-area bilayer graphene with rich rotation angles would be important for the investigation of angle-dependent physics and applications, which, however, is still challenging. Here, we demonstrate a chemical vapor deposition (CVD) approach for growing high-quality tBLG using a hetero-site nucleation strategy, which enables the nucleation of the second layer at a different site from that of the first layer. The fraction of tBLGs in bilayer graphene domains with twist angles ranging from 0° to 30° was found to be improved to 88%, which is significantly higher than those reported previously. The hetero-site nucleation behavior was carefully investigated using an isotope-labeling technique. Furthermore, the clear Moiré patterns and ultrahigh room-temperature carrier mobility of $68,000 \, \text{cm}^2 \, \text{V}^{-1} \, \text{s}^{-1}$ confirmed the high crystalline quality of our tBLG. Our study opens an avenue for the controllable growth of tBLGs for both fundamental research and practical applications.

---

[1] Center for Nanochemistry, Beijing Science and Engineering Center for Nanocarbons, Beijing National Laboratory for Molecular Sciences, College of Chemistry and Molecular Engineering, Peking University, Beijing 100871, People's Republic of China. [2] Academy for Advanced Interdisciplinary Studies, Peking University, Beijing 100871, People's Republic of China. [3] Beijing Graphene Institute, Beijing 100095, People's Republic of China. [4] School of Physics and Astronomy, University of Manchester, Manchester M13 9PL, UK. [5] Soochow Institute for Energy and Materials Innovation, Soochow University, Suzhou 215006, People's Republic of China. [6] Department of Modern Mechanics, University of Science and Technology of China, Hefei 230026, People's Republic of China. [7] Clarendon Laboratory, Department of Physics, University of Oxford, Oxford OX1 3PU, UK. [8] School of Physical Science and Technology, ShanghaiTech University, Shanghai 201210, People's Republic of China. [9] Hefei National Laboratory for Physical Sciences at the Microscale, University of Science and Technology of China, Hefei 230026, People's Republic of China. [10]These authors contributed equally: Luzhao Sun, Zihao Wang, Yuechen Wang. ✉email: hshnpu@ustc.edu.cn; hlpeng@pku.edu.cn; linli-cnc@pku.edu.cn; zfliu@pku.edu.cn

Recently, twisted bilayer graphene (tBLG), which is composed of two graphene layers with an interlayer twist angle ($\theta$)[1], has emerged as an exciting material for both fundamental studies and practical applications because of its unique $\theta$-dependent properties[2,3]. Specifically, the presence of twist angle causes van Hove singularities (vHSs) to emerge in the electronic density of states, resulting in an enhanced optical absorption[4] and photocurrent generation at certain wavelengths[5,6]. Furthermore, the realization of a correlated Mott-insulator and super-conductivity, at the magic angle (tBLG with $\theta$ of 1.1°), has attracted worldwide interest[7,8]. Consequently, an approach for producing large-area, high-quality tBLG, with a full range of $\theta$ from 0° to 30° would significantly improve the ability to investigate its unique physical properties and applications. To achieve this, artificially stacking methods have been developed, but suffers from unavoidable interlayer contamination which affects the interlayer coupling[6,9]. Prior investigations also indicated that azimuthally disordered multilayer graphene could be grown on C-terminated SiC[10]. Recently, 30°-tBLG was obtained by using a sacrificial hexagonal boron nitride (hBN) layer on Si-terminated SiC[11]. However, study and applications of the SiC-epitaxial tBLG are currently limited by the tedious transfer process and the relatively high price of SiC single crystal. In addition, the intermediate layer could also enable the formation of twist angle when growing the other two-dimensional (2D) materials[12].

Direct growth of bilayer graphene on transition metal substrates such as Cu[13–21] or Cu-Ni alloy[22,23] via chemical vapor deposition (CVD) is currently considered one of the most promising methods, due to the high scalability and the excellent quality of CVD graphene[24–26]. However, during the high-temperature CVD growth, the energetically favorable bilayer graphene structure is the non-twisted one, i.e., AB-stacked bilayer graphene (AB-BLG), whereas any rotation between the two layers would need to overcome a high energy barrier[27,28]. Thus, compared to AB-BLG, the fraction of tBLG were limited to around 50% in previous reports[14,17,19,29]. Thus, an efficient approach for growing tBLGs, especially in high fraction of twisted graphene and with a full range of twist angles, is still highly desirable.

In CVD approach, since the microscopic environment surrounding the nucleation site determines the orientation of graphene, two layers in bilayer graphene sharing the same nucleation center would preferentially grow with either the same orientation or solely with a twist angle of 30°. Therefore, in this study, a hetero-site nucleation strategy, where the two graphene layers nucleate at different sites, is developed to significantly enhance the fraction of tBLGs in bilayer graphene domains to 88%, with the range of $\theta$ from 0° to 30°. As proven by using a carbon-isotope-labeling technique, the gas-flow perturbation was capable of controllably initiating this hetero-site nucleation. The high crystalline quality of the as-grown tBLGs was confirmed by the clear Moiré patterns in high-resolution (HR)-transmission electron microscopy (TEM), and by the ultrahigh carrier mobility of 68,000 cm$^2$ V$^{-1}$ s$^{-1}$ at room temperature. This synthesis strategy by the intentional design of nucleation sites would bring more inspiration for the controlled growth of graphene and other 2D materials with precise control over interlayer twist in the near future.

## Results

**The hetero-site nucleation strategy.** During the high-temperature CVD process, high partial pressure of H$_2$ results in the H-terminated edges of graphene and therefore reduces interaction with Cu substrate, which promotes the diffusion of the carbon species beneath the first layer to fuel the growth of the second layer graphene[16,30]. In the common bilayer graphene growth results without introducing the gas-flow perturbation, the two graphene layers would share the same nucleation site and grow simultaneously[16]. In this case, the same surrounding microscopic environment, including Cu steps and particles, would result in the same crystalline orientation of the two layers,

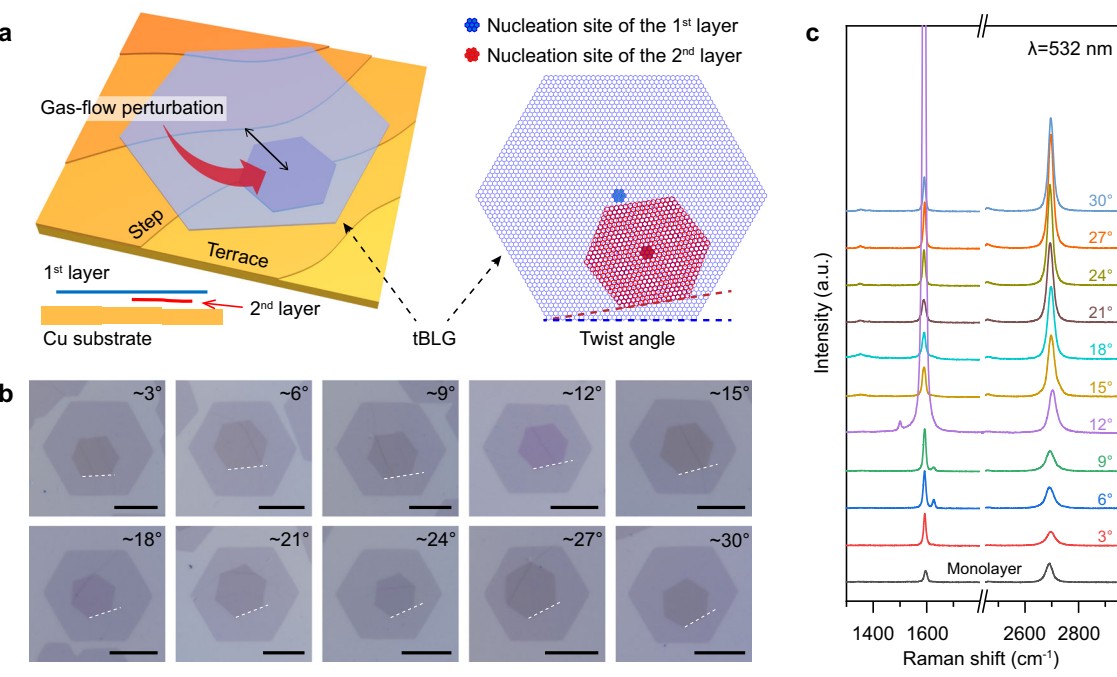

**Fig. 1 Hetero-site nucleation strategy for growing tBLG. a** Schematic of the hetero-site nucleation for growing tBLG on a Cu substrate, where the nucleation site of the second layer graphene (red) is different from that of the first layer (blue). The side view illustrates that the second layer graphene grows beneath the first layer. Note that the nucleation behavior of graphene is usually determined by microscopic environment surrounding the nucleation site, such as Cu steps. **b** OM images of as-grown tBLGs with twist angles of ~3°, ~6°, ~9°, ~12°, ~15°, ~18°, ~21°, ~24°, ~27°, and ~30°; scale bars: 10 µm. **c** Raman spectra of corresponding tBLGs samples in panel (**b**).

thus AB-BLG with no interlayer rotation is preferentially formed (Supplementary Fig. 1a). In contrast, after nucleation of the first graphene layer, subsequent nucleation of the second graphene layer is initiated by introducing a gas-flow perturbation, and therefore the nucleation of the second layer occurs at a distinct site, i.e., hetero-site nucleation (Fig. 1a). Therefore, the orientation of new layer would be determined by a different local environment, enabling the presence of interlayer twist and the formation of tBLGs. A sudden increase of $H_2$ and $CH_4$ is introduced here as a gas-flow perturbation, which would provide more active hydrogen and active carbon species to fuel the nucleation and growth of the second layer graphene (Supplementary Fig. 1b). Note that the sudden carbon-source supply enhancement is previously reported to be capable of initiating new nucleation of monolayer graphene (MLG)[31,32]. Optical microscopy (OM) was used to measure the resulting $\theta$ of the tBLG, based on the sharp edges of the hexagonal tBLG domains (Fig. 1b), which clearly indicates that tBLGs with a wide range of twist angles were grown successfully. Since the centers of hexagonal graphene domains are typically the original nucleation sites of the layers[19], the non-concentric structure of the tBLG domains observed in Fig. 1b clearly confirm the hetero-site nucleation behavior of the second layer graphene.

Raman spectra were acquired to characterize their crystalline quality and interlayer coupling of the as-grown tBLGs, which exhibit strong $\theta$-dependent vibrational modes (Fig. 1c and Supplementary Fig. 2). At a low twist angle (<10°), an R′ band is clearly observed, while the 2D band intensity is relatively weak compared to that of MLG, consistent with the reported results of tBLGs[33]. In addition, The G band is strongly enhanced at a twist angle of ~12°, because the incident photon energy ($\lambda = 532$ nm, $E_{ex} = 2.33$ eV) matches the energy between vHSs of the tBLG[34].

**Mechanism of the hetero-site nucleation.** Carbon-isotope-labeling experiments in conjunction with Raman spectroscopy were performed to investigate the hetero-site nucleation growth mechanism. Here, $^{12}C$-labeled and $^{13}C$-labeled methane ($^{12}CH_4$/ $^{13}CH_4$) were sequentially introduced to the CVD chamber in alternating 5 min periods over a total of four cycles. To confirm its contribution to the hetero-site nucleation behavior, the flow perturbation was introduced, by increasing the flow rates of $H_2$ and $CH_4$, after either 5 or 10 min (Fig. 2a, b). The nucleation centers of each layer can be inferred from the hexagonal domain shapes observed in the OM images (Fig. 2c, d). When flow perturbation was introduced at 5 min, a small shift in the nucleation center of the new layer, from the original, is observed (Fig. 2c). This shift increases when perturbation was introduced after a longer duration of 10 min (Fig. 2b, d). Furthermore, because the first and second layers both follow a surface-mediated growth mechanism, the spatial distribution of $^{12}C$ and $^{13}C$ in each layer can be visualized by Raman mapping, based on the different modes of $^{12}C$-graphene and $^{13}C$-graphene (Supplementary Fig. 3)[20,35]. In the Raman 2D$^{13}$-intensity maps of the as-grown tBLGs, the nucleation center shifts in the second layer can be clearly visualized by the isotope distribution (Fig. 2e, f and Supplementary Fig. 4). In general, the first graphene layer displays four cycles of alternating $^{12}CH_4$ and $^{13}CH_4$, indicating the growth duration of 20 min. In contrast, when perturbation was introduced at 5 or 10 min, the second graphene layer exhibits three or two cycles, confirming the growth duration for second layer is 15 or 10 min, and their domain centers are located right at the $^{12}C$-$^{13}C$ boundary of the first layer. These results confirm that the second layer nucleation occurs at either 5 or 10 min, exactly when perturbation was introduced. In addition, plotting the time evolution of the two graphene layers (Fig. 2g) can also reveal the

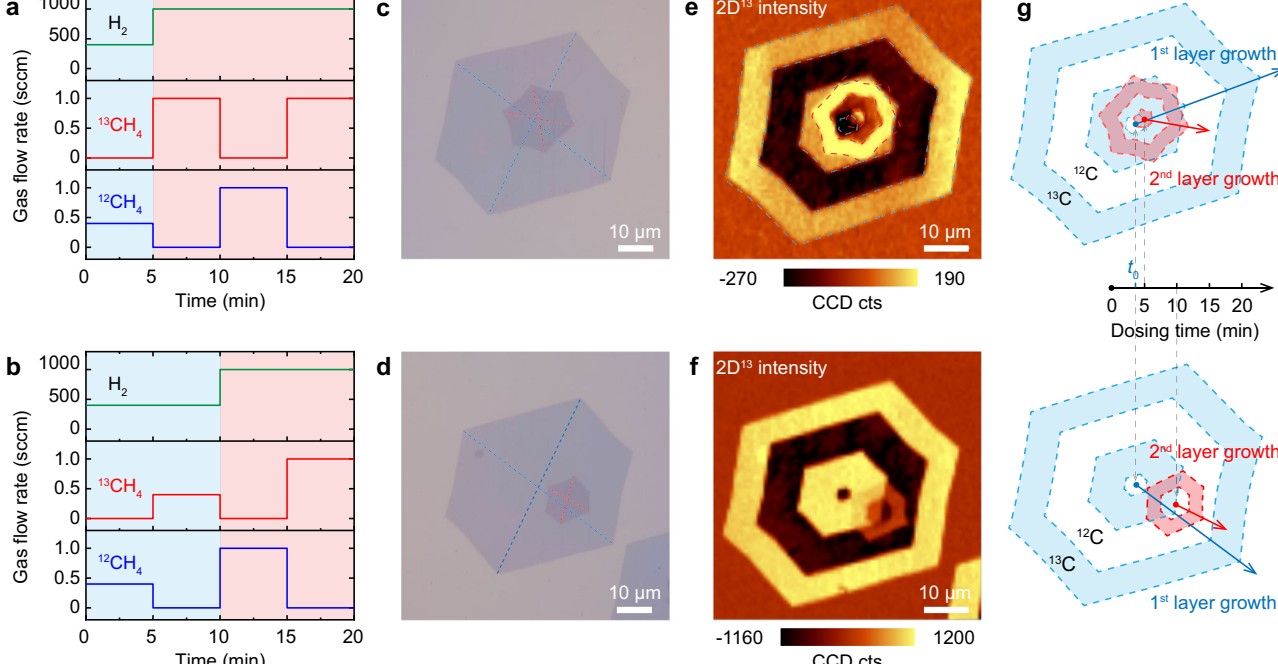

**Fig. 2 Hetero-site nucleation and the growth process of tBLGs visualized by isotopic labeling in conjunction with micro-Raman spectroscopy.**
**a**, **b** Feedstock feeding process for growth of isotope-labeled tBLG using the hetero-site nucleation strategy, where gas-flow perturbation is introduced at 5 min (**a**) and 10 min (**b**), respectively. **c**, **d** Resulting OM images of transferred tBLGs. The dashed lines guide the eye to the nucleation sites of the first and second graphene layers, represented by the intersections of blue and red lines, respectively. **e**, **f** Raman 2D$^{13}$-intensity maps (integrated from 2560 to 2620 cm$^{-1}$) of areas corresponding to (**c**) and (**d**), respectively. **g** Schematics of the isotopic distribution and corresponding growth processes of the tBLGs corresponding to (**e**, top) and (**f**, bottom), respectively. The blue and red rings represent the first and the second graphene layers, respectively. The time axis shows the nucleation times of the second layers, which can be controlled by the introduction of gas-flow perturbation.

nucleation time for the second layer, which can be obtained by the intersection of the fitted growth line and the time axis (Supplementary Fig. 4b, d)[32]. Thus, we can confidently conclude that the hetero-site nucleation of the second layer was controllably induced by the introduction of perturbation. In addition, a nucleation time ($t_0$) is required for the first layer, after dosing of $^{13}CH_4$ or $^{12}CH_4$, because it must overcome an energy barrier to form a stable nucleus[32]. The twist angles of tBLGs in Fig. 2c, d are ~30° and ~9°, respectively, which are remarkably different from the $^{12}C$-$^{13}C$-labeled AB-BLG grown without flow perturbation (Supplementary Fig. 5), confirming the importance of hetero-site nucleation for growing tBLGs.

Computational fluid dynamics (CFD) simulations were conducted to investigate the dynamics of the gas flow during the perturbation. After the sudden increase of flow rate of $H_2$ and $CH_4$, the gas-flow velocity increases rapidly at first, and return to the original value within 5 s (Supplementary Fig. 6a, b). Note that the gas-flow velocity in the boundary layer is slow; therefore, the influence of the fluctuation of the gas-flow velocity near the sample surface on the nucleation of the second layer graphene is really limited. A uniform distribution of pressure across the entire the tube reactor is observed, and the uniformity retains after the increasing of the pressure caused by the perturbation. Because of the feedback of chamber pressure—pumping rate (mass flow rate) —chamber pressure, the increasing rate of the pressure would decrease gradually with the time: the real-time pressure reaches 80% of the target value within 5 s and finally reaches a new steady value in about 20 s (Supplementary Fig. 6e, f). The concentrations of $H_2$, $CH_4$, active hydrogen, and active carbon species also increase accordingly (Supplementary Fig. 7).

Generally, when the concentration of active carbon species is enhanced, the nucleation rate and the growth rate would be increased[31,32,36]. However, because the partial pressure of $H_2$ is greatly enhanced simultaneously, the graphene edges would be terminated by hydrogen, which hinders the attachment of active carbon species to the edge of the first layer[32,37]. According to the theoretical calculation, only carbon atoms can diffuse under the first layer to fuel the growth of the second layer graphene[20,30]. Thus, if the nucleation of the second layer graphene occurs at the same site with that of the first layer, several energy barriers must be overcome (Supplementary Fig. 8): (1) diffusion barrier-I corresponding to energy barrier needed to be overcome when carbon atoms diffuse from the bare Cu surface to graphene-covered region; (2) diffusion barrier-II needed to be overcome when carbon atoms diffuse beneath the first layer graphene; and (3) nucleation barrier of the second layer graphene. The diffusion barrier-I, which is previously calculated to be ~0.6 eV[30], will

impede the diffusion of carbon atom toward the graphene-covered region, and increase the concentration of carbon atoms near the edge of first layer graphene, which in turn promote the nucleation near the graphene edge. Near the edge of the first layer, the high-energy sites (active sites) of the substrate in the presence of steps, kinks, and particles would capture the C atoms and result in the nucleation of the second layer, consistent with the reported literature[38]. Therefore, when gas-flow perturbation is introduced, the second layer preferentially nucleates near the edge of the first layer, rather than at the original nucleation center, and the local environments near the nucleation sites of the two graphene layers are therefore different, which is responsible for the formation of interlayer twist (Supplementary Fig. 9). In addition, the nucleation site of the second layer graphene can be controlled by the timing of the flow perturbation introduction, which agrees with the results in Fig. 2g.

**Key parameters for growing tBLG.** To achieve hetero-site nucleation for tBLG growth, three key conditions are required: (1) no second layer can form during the nucleation stage of the first layer; (2) the second hetero-site nucleation must be triggered by a perturbation; and (3) carbon source must be sufficient for the growth of second layer. A series of experiments were conducted to improve the growth controllability with respect to the three conditions. The ratio of MLG domains (second-layer-free domains) to all graphene domains ($R_{MLG}$) as functions of the $H_2$ and $CH_4$ (carbon-source supply) flow rates reveal the presence of two regions, as separated by green dash line (Fig. 3a and Supplementary Fig. 10a, b). One region (left bottom) consists of the corresponding parameters ($H_2$ and $CH_4$ flow rates) that are suitable for growing second-layer-free graphene domains, which was chosen during the first nucleation step (step I). The other region (top right) is composed of the parameters for growing second-layer-rich graphene domains, which was used during the second layer growth step (step II).

In the hetero-site nucleation process, the increase of both $H_2$ and $CH_4$ flow rates are important for seeding and growth of the second layer graphene. The ratio of bilayer graphene domains to all graphene domains ($R_{BLG}$) as a function of the ratio of $H_2$ flow rate in step II to that in step I ($H_{2\ \text{step II/step I}}$) is displayed in Fig. 3b. Clearly, a higher $H_{2\ \text{step II/step I}}$ in the flow perturbation is beneficial for increasing $R_{BLG}$, highlighting the importance of increase the $H_2$ partial pressure in step II. High $H_2$ partial pressure enables H-termination of the graphene edge, which allows more active carbon species to diffuse beneath the first layer of graphene to fuel the growth of the second layer graphene[30]. Meanwhile, at the same partial pressure of hydrogen,

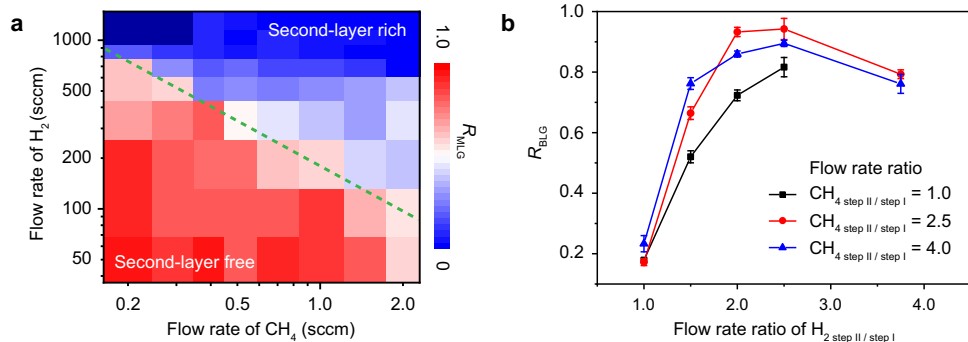

**Fig. 3 Key parameters for growing tBLGs. a** $R_{MLG}$ as functions of the gas-flow rates of $H_2$ and $CH_4$, respectively. The red and blue blocks denote the second-layer-free and second-layer-rich regions, respectively. The horizontal and vertical coordinates are logarithmic, while the color varies linearly with the ratio of second-layer-free domain to all graphene domains. **b** $R_{BLG}$ as functions of the flow rate ratio of $H_{2\ \text{step II/step I}}$. Black, red, and blue curves represent different $CH_4$ flow rate ratio ($CH_{4\ \text{step II/step I}}$) of 1.0, 2.5, and 4.0, respectively.

$R_{BLG}$ would be higher when the flow rate of $CH_4$ in step II is higher than that in step I (Fig. 3b). However, an excess partial pressure of $H_2$ or $CH_4$ (over 1200 and 1.5 sccm for $H_2$ and $CH_4$, respectively) would induce undesirable few-layer graphene (FLG) formation, and therefore slightly reduce the $R_{BLG}$ (Supplementary Fig. 10c, d). Therefore, the flow rate ratios of $H_{2\ step\ II/step\ I}$ and $CH_{4\ step\ II/step\ I}$ must be carefully controlled in the hetero-site nucleation step (Fig. 3b). By measuring the edge direction of as-grown BLG domains (Supplementary Table 1 and Supplementary Figs. 11 and 12) in conjunction with the Raman spectroscopy (Supplementary Fig. 13 and 14), the ratios of tBLG the corresponding twist-angle distribution can be obtained. Without hetero-site nucleation, the twisted fraction in all the bilayer domains is only 16%, while it is 86% when hetero-site nucleation was employed (Supplementary Fig. 15). This increase highlights the importance of our hetero-site nucleation strategy for growing tBLG. Furthermore, a wide distribution of twist angles were observed, by measuring the sharp edges of the hexagonal tBLG domains in OM images, with a relatively higher fraction of bilayers with twist angles around 0° and 30° (Supplementary Fig. 15d)[17].

**TEM characterization of tBLGs.** The twist angles of the as-grown tBLGs were further characterized by TEM and selective-area electron diffraction (SAED). Note that, for a vivid evaluation, SAED was conducted on suspended graphene, spread over 600 holes of Quantifoil substrate (Supplementary Fig. S16). Figure 4a shows a representative tBLG SAED pattern, exhibiting two groups of hexagonal points with a relative rotation of 9°, which reflects the twist angle between the two layers. Because of the relatively weak interactions between the two layers, the corresponding intensity ratio of the diffraction points ($I_{\{2100\}}/I_{\{1100\}}$) is lower than unity (Fig. 4b), which is consistent with previous tBLG results[17]. The fraction of tBLG within all bilayer domains, and the corresponding distribution of twist angles were obtained by analyzing all the obtained diffraction patterns (Fig. 4c, Supplementary Figs. 17–20). Interestingly, the fraction of tBLG was calculated to be as high as 88%, which represents a striking increase over the results obtained without the hetero-site nucleation strategy[17]. Note that tBLG with very small twist angle (less than 3°) is difficult to grow because of the relatively higher stability of AB-BLG and[27]. The fraction of twist angles near 30° is significantly higher than the other tBLGs.

Figure 4d–g shows HR-TEM images of representative tBLGs with twist angles of 2.9°, 9.4°, 18.3°, and 30°, respectively, all of which show clear tBLG super-lattices that confirm the high crystalline quality. In addition, the corresponding Moiré patterns, with a Moiré period of ~4.9 nm, are clearly observed for the 2.9° tBLG. The signature 12-fold rotational symmetry is also clearly visible in the as-grown 30° tBLG, where the Stampfli tiles, including equilateral triangles, squares, and rhombuses can fill the entire space with different orientations (Supplementary Fig. 21)[11,12]. The rotational order, but lack of translational symmetry, indicates that a high-quality quasi-crystalline system was formed.

**Electronic quality of tBLGs.** An hBN-encapsulated tBLG Hall bar device, with one-dimensional edge contact, was fabricated to investigate the electronic quality of as-grown tBLGs (Fig. 5a). Notably, it is difficult to pick CVD-grown tBLG directly up from the Cu substrate[39,40], because of the relatively weak interlayer interactions. Thus, the as-grown tBLGs were first transferred onto a $SiO_2/Si$ substrate with the assistance of poly(methyl methacrylate) (PMMA). The selected tBLG domain was then picked up from the $SiO_2$ substrate using an hBN flake, denoted as top hBN or t-hBN, and subsequently dropped down onto another hBN flake, denoted as bottom hBN or b-hBN, on $SiO_2$, to form an hBN/tBLG/hBN sandwich structure (Supplementary Fig. 22)[41]. Annealing was then conducted to help clean the interfaces in the

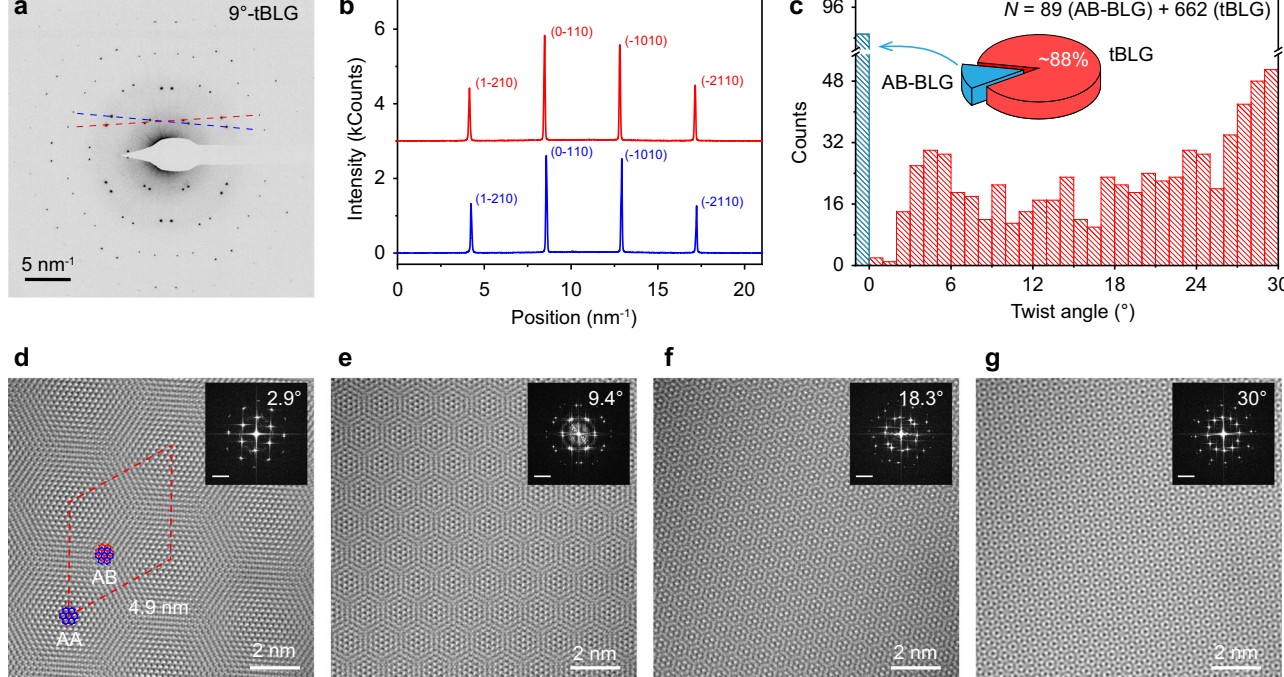

**Fig. 4 TEM characterization of as-grown tBLG. a** Typical SAED pattern of tBLG with a twist angle of ~9°. **b** Intensity profiles, along the axes marked in (**a**) with red and blue dashed lines. **c** Statistical results of stacking order (AB stacking or non-AB stacking) and distribution of twist angles based on SAED patterns of as-grown bilayer graphene domains. **d–g** HR-TEM images of tBLG with clear Moiré patterns. Insets: Fast Fourier transforms (FFTs) of the corresponding HR-TEM images; scale bars: 5 nm$^{-1}$.

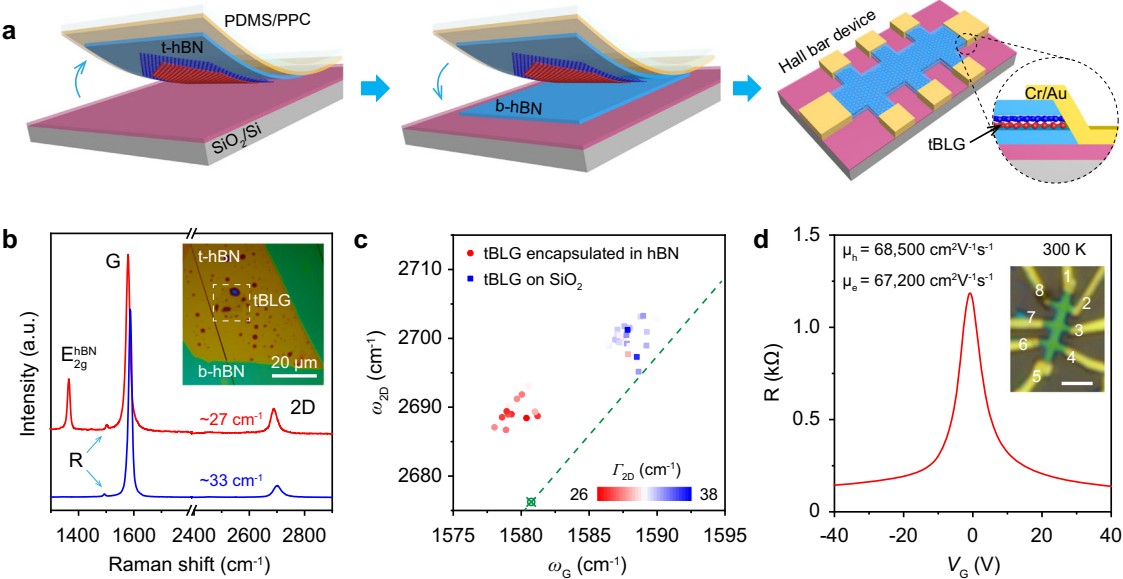

**Fig. 5 Ultrahigh carrier mobility of as-grown tBLG. a** Schematic of the fabrication process of hBN-encapsulated Hall bar devices with edge-contact between tBLG and Cr/Au electrodes. **b** Raman spectra of ~12° tBLG encapsulated in hBN (red curve), and on SiO₂ (blue curve). Inset: enhanced OM image of ~12° tBLG encapsulated in hBN. **c** Correlation plot of 2D peak position ($\omega_{2D}$) as a function of G peak position ($\omega_G$). The circles and squares represent the data taken from tBLG encapsulated in hBN and on SiO₂, respectively. The colors of all the data points represent the full-width at half-maximum (FWHM) of the 2D peak, $\Gamma_{2D}$, according to the color scale bar included in the figure. **d** Four-terminal resistance as a function of gate voltage ($V_G$) at room temperature ($T = 300$ K). Inset: OM image of the encapsulated tBLG Hall bar device, scale bar: 2 μm.

hBN/tBLG/hBN structure, yielding blister-free regions for device fabrication (Supplementary Fig. 23)[42,43].

Raman spectroscopy is sensitive to global environmental factors such as doping, strain, and flatness[44,45]. For the tBLG sample on SiO₂ (blue curve in Fig. 5b), the G peak position ($\omega_G$) is ~1587 cm⁻¹, with a full-width at half-maximum (FWHM) of the G peak ($\Gamma_G$) of 11 cm⁻¹, while the 2D peak position ($\omega_{2D}$) is ~2700 cm⁻¹, with the FWHM ($\Gamma_{2D}$) of 34 cm⁻¹. The high $I_G/I_{2D}$ peak ratio of ~20, together with the R peak at 1493 cm⁻¹, indicate an interlayer twist angle of ~12° (laser wavelength $\lambda = 532$ nm). For the tBLG encapsulated in hBN (red curve in Fig. 5b), the $I_G/I_{2D}$ peak ratio decreased to ~7; the $\omega_G$ and $\omega_{2D}$ showed clear red shifts to ~1580 and ~2689 cm⁻¹, respectively; and the $\Gamma_{2D}$ decreased to ~27 cm⁻¹, all of which can be attributed to the fine encapsulation (Fig. 5b, c)[43,44]. Electrical transport measurements were conducted to probe the quality of the tBLG with a twist angle of 12°. Figure 5d shows the resistivity as a function of gate voltage $V_G$ at room temperature. By plotting the conductivity as a function of the carrier density ($n$) and performing a linear fit near the charge neutrality point (Supplementary Fig. 24), we obtain the room-temperature carrier mobility of 67,000 and 68,000 cm² V⁻¹ s⁻¹ for electrons and holes, respectively, which confirm the high quality of the as-grown tBLG[41]. We further investigated the electronic band structures of as-grown tBLGs (~3°, ~6°, and ~11°) by using the angle-resolved photoemission electron spectroscopy with submicrometer spatial resolution (micro-ARPES). The $\theta$-dependent vHSs are clearly observed in the energy–momentum–dispersion diagrams along with the corresponding integrated energy distribution curves (EDCs) (Supplementary Fig. 25).

## Discussion

This work describes a hetero-site nucleation strategy for successfully growing the tBLG with the fraction of tBLG as high as 88%, and a wide range of twist angle from 0° to 30°. By employing the gas-flow perturbation after the nucleation of the first layer, the second layer nucleation could be initiated at a different location, which was investigated by isotope labeling in conjunction with

micro-Raman spectroscopy. The sudden increase of carbon-source supply would drive the system out of equilibrium, and allow the formation of the second-layer nuclei without the pre-ferred equilibrium stacking order. In this work, since the second layer graphene locates underneath the first layer, substrate plays crucial roles in determining the crystal orientation of graphene. The interaction between Cu substrate and the second layer gra-phene is much stronger than the interlayer interaction of bilayer graphene during the nucleation stage, which would suppress the equilibrium of AB-stacking configuration[46,47]. The steps, kinks, and particles on Cu surface would provide different chemical environment near the second-layer nucleation sites, which is mainly responsible for the formation of interlayer twist. The high quality of the as-grown tBLG was corroborated by clear Moiré patterns in the HR-TEM and carrier mobility exceeding 68,000 cm² V⁻¹ s⁻¹ at room temperature. Our work brings inspiration for controlled growth of graphene and other 2D materials with interlayer twist, and future works still need to be done to grow tBLGs with certain twist angles, possibly by con-trolling the nucleation density, designing the growth substrate[48], utilizing the axial screw dislocation[49,50], or introducing inter-mediate layers[11,12]. Meanwhile, the torque toward smaller twist angles, which increases rapidly near the zero-angle, should be overcome to improve the portion of very small twist angle (near magic angle).

## Methods

**tBLG growth.** tBLGs were grown on commercially available 50 μm thick Cu foil (Kunshan Luzhifa Electron Technology Co., Ltd., China) in a low-pressure CVD system. Cu-foil pieces were placed in a quartz-tube furnace (Tianjin Zhonghuan Furnace Corporation, SK-G15123K-3-940) and sequentially heated to 800 °C for 30 min (500 sccm Ar), annealed for 10 min at 800 °C (500 sccm Ar), heated to 1020 °C for 10 min (500 sccm H₂), and annealed for 30 min at 1020 °C (500 sccm H₂). Graphene growth proceeded by introducing ¹²CH₄ or ¹³CH₄ (99% purity, Sigma-Aldrich) after the flow of H₂ was stable at the appropriate value (parameters are shown in Figs. 2 and 3 and Supplementary Figs. 5 and 10). Note that the relationship between partial pressure (Pa) and flow rate (sccm) of H₂ in our CVD system can be approximately described by the formula: $P \approx 0.9f$, where $P$ is the partial pressure, and $f$ is the flow rate.

**Graphene transfer**. tBLGs were transferred onto SiO$_2$ with the assistance of PMMA. The tBLG/Cu sample was spin-coated with PMMA (2000 rpm) and baked at 170 °C (3 min), followed by the removal of the Cu foil by etching in 1 M Na$_2$S$_2$O$_8$ solution. After being washed with deionized water, the PMMA/graphene was subsequently placed onto SiO$_2$ and the PMMA was dissolved with acetone. The graphene was transferred onto a TEM grid using a non-polymer-assisted method, as reported previously[51].

**Characterizing the tBLG samples**. OM images of tBLGs on SiO$_2$/Si substrates (SiO$_2$ thickness: 90 or 285 nm) were obtained by OM (Nikon LV100ND equipped with DS-Ri2 camera). Raman spectra and maps were obtained using a Horiba HR800 instrument with a 532 nm laser or a Witech Alpha RSA300+ instrument with a 488 nm laser. TEM images, SAED patterns and Moiré pattern images were collected using the instrument of FEI Tecnai F30 (at 300 keV electron energy) and an aberration-corrected TEM (FEI Titan Cubed Themis G2 300; at 80 keV electron energy). SEM images were acquired on the instrument of Hitachi S-4800 and Thermo Scientific Quattro S (at 2 keV electron energy). AFM images of tBLG on Cu foil is characterized by using Bruker Dimension Icon with ScanAsyst mode. ARPES experiment was performed on the Spectromicroscopy beamline, Elettra Synchrotron (Italy). Before the measurement, tBLG samples was in situ annealed at 300 °C for 3 h. The measurement was performed in the high vacuum (better than 5 × 10-10 mbar) at 80 K. All the data were collected with an incident photon energy of 74 eV.

**hBN-encapsulated tBLG device fabrication**. An hBN flake was picked up at 55 °C by a PPC/ PDMS stack on a glass slide, which was attached to a micromanipulator. The as-formed hBN/PPC/PDMS stack was then used to pick up the tBLG from the SiO$_2$/Si at 55 °C, because the van der Waals forces between hBN and the tBLG are relatively stronger than those between SiO$_2$ and the tBLG. Consequently, the tBLG/hBN/PPC/PDMS stack was brought into contact with another hBN flake, after which the tBLG/hBN was released from the PPC at 70 °C, yielding an hBN/tBLG/hBN heterostructure. Annealing (350 °C in air) was necessary following the construction of hBN/tBLG/hBN, which helped to clean the interfaces between hBN and tBLG[41–43]. The contamination might come from the transfer step and high-temperature growth step[52], thus, using the strategies for growing clean tBLGs or no polymer transfer would contribute to the removal of contamination. The Hall bar device was fabricated with alignments marks. To pattern graphene into a Hall bar geometry, electron-beam lithography and reactive ion etching (RIE) were employed. Cr/Au (3/50 nm) electrodes were deposited by electron-beam evaporation.

**CFD simulation**. The CFD model here is based on laminar compressible flow with surface chemical reaction. Settings on gas mixture, chemical reactions are based on the previous reported literature[53]. CFD code ANSYS FLUENT is used here, and the computational mesh arrangement of over 200,000 cells is developed with fine mesh near the sample surface to guarantee the convergence. The geometry and boundary conditions of gas inlet and temperature are set as shown in Supplementary Fig. 27. The outlet boundary condition is based on the performance curve of the mechanical pump in our experiment (ULVAC, GCD-136X).

## Data availability

The authors declare that the data supporting this study are available within the article and its Supplementary Information files. Further information is also available from the corresponding authors upon reasonable request. Source data are provided with this paper.

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

## Acknowledgements

We thank Prof. Qinghong Yuan in East China Normal University for the theoretical discussion. We thank Hailong Hu in WITec GmbH Beijing Representative Office for the help on the processing of Raman data. This work was financially supported by Beijing National Laboratory for Molecular Science (BNLMS-CXTD-202001), the Beijing Municipal Science & Technology Commission (No. Z181100004818001, Z191100000819005, and Z201100008720005), the National Basic Research Program of China (No. 2016YFA0200101 and 2016YFA0200103), the National Natural Science Foundation of China (NSFC, No. 21525310, 51432002, 51672181, and 21825302), and Major International (Regional) Joint Research Project (No. 21920102004). The ARPES experiment was supported by the beamline scientists in Spectromicroscopy Elettra under the proposal No. 20195301. D.P. thanks the support from China Scholarship Council.

## Author contributions

Z. Liu, L.L., H.P., and L.S. conceived the experiment. L.S. and Y.W. conducted the growth, transfer and optical microscopy characterization of tBLGs. L.Z., L.S., and M.H.R. conducted the TEM experiment. L.S., Y.W., and L.L. conducted the error analysis of the twist-angle measurements. Y.W., Y.L., Z. Li, B.C., L.S., and S. Zhong did the statistics of twist angles. K.S.N., L.L., and W.W. fabricated the hBN-encapsulated tBLG devices. K.S.N. and Z.W. performed the electrical measurements. L.S., Y.W., S. Zhang, and L.T. conducted the Raman characterization. S.H. and Z.L. performed the CFD simulations. Y.C., D.P., and H.F. conducted the ARPES experiment. All authors discussed the results and wrote the manuscript.

## Competing interests

The authors declare no competing interests.
