## [Peer Review File · Nature Communications]

REVIEWER COMMENTS

Reviewer #1 (Remarks to the Author):

The manuscript reports an epitaxial growth method to grow tBLG on Cu foils. They can increase the ratio of tBLG to 86% (of total BLG?). They also used isotope to reveal the growth mechanism, and they attribute the formation of tBLG to the "turbulence" when changing gas-flow during the growth. The primary growth method has been reported in ref. 27 and 28. What different is the "turbulence". The method itself is not novel enough. Some of the evidences in this manuscript are not solid. My comments are as follows:

1. In the abstract lines 20-21, the author wrote: "Twisted bilayer graphene (tBLG) has recently attracted growing interest due to its unique twist-angle-dependent electronic properties. Fundamental investigations and applications of tBLG rely heavily on the availability of large-area tBLG with a broad range of twist angles." In the whole manuscript, there is no result related to "twist-angle-dependent electronic properties"; There is no demonstration that this method can benefit the "Fundamental investigations and applications". It needs more to show the potential, for example, how this growth method can benefit the field of twistrionics?
2. In the abstract lines 23-24, the author wrote: "...current fabrication methods are hindered by the small twisted region size or the large portions of non-twisted bilayer graphene...". The mainstream to produce precisely twist angle control MLG samples is to use the "dry" transfer method to pick up part of exfoliated graphene then twist stack it on the remnant graphene. Please address where the "large portions of non-twisted bilayer graphene" comes from.
3. In the abstract line 27-29, "The portion of tBLGs with twist angles ranging from 0° to 30° were found to be improved to 86%, which is significantly higher than those reported previously." According to the data, I think 86% means 86% of total BLG counts are tBLG. Please make it clear in whole manuscript.
4. In the abstract line 32-33, "Our study opens a new avenue for the controllable growth of tBLGs for both fundamental research and practical applications." According to the data in the manuscript, I do not think it is a "controllable growth of tBLGs", neither to achieve a precisely twist angle control or a large scale.
5. Line 75, "...by introducing a gas-flow turbulence...". The author should explain the mechanism of the "turbulence". By changing the gas-flow, both pressure, temperature, gas concentration, and the ratio will change. If there is no information about what is truly essential in the secondary nucleation, other researcher can't repeat the experiment. Because different CVD systems will have a different process of "turbulence". Line 123-132, do the author has any evidence or citation of this hypothesis? I suggest to do some simulation to support both "turbulence" and the hypothesis.
6. Line 80-82 and Fig. 3d. Twistrionics is sensitive to twist angle. I do not think the edges of graphene domains can be used to precisely determine the twist angle. Fig. 1b clearly shows that none of the graphene domains is a perfect hexagon. Some of them even have clearly nonparallel opposite sides. How did the author determinate 0 degree? If they cannot, where this "86%" comes from?
7. Fig. 3c should be together with their statistical distributions. Now it is meaningless.
8. Line 173-175, do the author have evidence or citations? Why in Fig. S11, there are some small twist angle samples?
9. Line 183-207, again, if there are no twist angles related to electrical properties, it is meaningless.

Reviewer #2 (Remarks to the Author):

This submission to Nature Communications presents a novel technique for growing twisted bilayer graphene over a wide range of twist angles. Both comprehensive growth methods and characterization (primarily Raman, TEM and electrical mobility) confirm the utility and promise of the new 'ex-situ' growth approach presented.

CVD grown twisted bilayer graphene is common, but the current methods favouring tBLG growth are haphazard at best. Numerous groups cited have already achieved CVD growth of bilayer graphene where >50% of bilayers twisted (e.g. Hone, Park, and many others, as confirmed by darkfield TEM) for nearly a decade. However, all CVD growth that I am aware seem largely random occurrences despite what the literature may claim. Typically, subtle changes in foil-selection or differing furnace result in entirely different growth patterns of the bilayer. Most often, the second layer is either all Bernal (AB stacked) or the angle distribution is poor. This work appears to break this 'trial-and-error' approach to CVD tBLG growth by developing a second layer induced nucleation trigger by induced turbulence of furnace airflow. This is best evidenced by comparing Fig. 3d and 4c that show the AB growth are reduced from 85 to 15%. I am aware of papers that claim up to 50% tBLG over AB stacked, but 85% is a new achievement for CVD growths (to my knowledge). This novel approach merits broad communication given the recent topical interest in tBLG (owing to its to unexpected superconducting and Mott insulator properties seen in exfoliated samples).

As someone who does not grow CVD graphene, I can only comment on the utility to the experimental community and the quality of characterization methods employed. I believe after minor revision both are sufficiently high to merit publication in Nature Communications. Nonetheless, reproducibility of the growth method proposed on alternate setups remains a concern. Ultimately I must yield to CVD growth experts to better access the novelty and reproducibility of this triggered second-layer growth mechanism.

The quality of data presentation and range of twist angles is achieved is a sufficiently exciting advance to me. I remain confident that most of the novel many-body electronic properties observed in exfoliated graphene will be reproduced in CVD samples such as the growths outlined in this paper. I hope this work will accelerate such discoveries.

Some requested revisions:

1) My primary concern remains that this paper does not characterize the twist angles with the accuracy needed for the broad community intended. Throughout they only characterize tBLG with an approximate angle (e.g. ~ 6 degrees). However, both TEM and optical methods can achieve far more precise angle characterizations, often to the first decimal point.

What is the error bar on the angle characterization? Can graphs such as Fig. 3d and 4c be binned smaller than 2-degree increments? This would be especially helpful in the 0-2 degree bin. To attract broad interest this paper needs more convincing data that <3 degree tBLG growths are likely by this method. Otherwise, the intended audience will not be convinced that 'magic angle' flatband electronic behaviour is promising from this CVD growth method. More careful aperture and apertureless TEM (esp. for some lower angle cases) is strongly recommended.

2) Comparing Fig. 3d and 4c, I may have missed a comparison of what % of the overall growth areas were single layer vs. bilayer. Systematic minimization of AB growth is important, but only if it does not substantially reduce the overall prevalence of bilayer vs. single layer coverage.

3) Searching for information on intrinsic doping and/or Fermi level characterization of the growth came-up absent. As Hall-bar conductance is presented in Fig. 5b, why not estimate the Fermi-level of the sample shown and comment more on the tBLG doping level generally. Hysteresis in RT conductance measurements (eliminated by cooling) can make this difficult, but rough estimates should be possible (and helpful).

Reviewer #3 (Remarks to the Author):

This manuscript reports the CVD growth of twisted bilayer graphene with a range of interlayer twist angles. Twisted bilayer graphene has excited a lot of interest recently, and the development of bottom-up strategies for growing bilayer graphene (as well as other 2D materials) with controlled twist angles would be an important step forward in this field. Here, the authors report a first step, namely overcoming the preferred (equilibrium) stacking of the second layer during growth. However, clearly several challenges remain. For example, the twist angles obtained are random and cannot be selected at this point. And very small interlayer twists, where much of the exciting many-body physics is observed, are not achievable since the second layer nucleus would likely snap into registry with the first layer. Nevertheless, the study is quite carefully conducted and should be of interest to experts in the field, and can therefore be considered for publication in Nature Communications. However, the paper contains a number of weak points and issues that are mandatory to be addressed conclusively before acceptance may be considered. A list of the most urgent issues follows below:

1) Title, abstract, and text: The term "ex-situ nucleation" is unfortunate, since "ex-situ" implies outside of the growth system. This will certainly lead to misunderstandings, and could negatively impact the paper. I strongly urge the authors to find a better term for their approach, which should emphasize the main characteristics, namely that nucleation is forced by a gas pressure spike. Perhaps "far from equilibrium" could be used.

2) Page 3, top: In the discussion of ways to overcome the equilibrium stacking registry, other approaches need to be included, even if they involve different 2D crystals other than graphene. This should include Ahn et al., Science 361, 782 (2018), where a sacrificial hBN layer on SiC was used to obtain 30 deg twisted bilayer graphene. And the discussion needs to include Sutter et al., Nature Commun. 10, 5528 (2019), where nucleation and transformation of an intermediate 2D crystal was used to obtain 30 deg twisted SnS₂.

3) The manuscript mentions "gas-flow turbulence" on several occasions, but there is no evidence that turbulence is indeed involved in the second-layer nucleation. Instead, it appears more likely just to be an effect of the rapid increase in the carbon supersaturation at higher precursor pressure, which forces the nucleation and does not allow it to take place in the otherwise preferred center of the flake. It appears that the gas spike drives the system far enough out of equilibrium that new nucleus does not adopt the preferred equilibrium stacking but grows with random orientation relative to the first graphene sheet. The authors need to carefully reconsider their explanation, especially a scenario that is based on non-equilibrium (kinetic) phenomena rather than "turbulence". If they believe that the underlying cause is indeed gas turbulence (in the strict hydrodynamic sense), then evidence has to be provided supporting this conclusion.

4) In the introduction and discussion, the authors need to clearly state the capabilities and limitations of the discussed method. While the CVD method can produce bilayer graphene with random twist angles, it is not able to reliably make bilayers with small twist angles (e.g., near the magic angle) and it is not (yet?) able to control the twist. This needs to be discussed especially carefully in the Discussion section, to inspire follow-up work addressing these limitations. Also, in the discussion, to contrast with their results the authors should discuss and cite competing approaches aimed at bottom-up synthesis of interfaces with small twist angles, e.g., using Eshelby twist in nanowires (Nature 570, 354 (2019), and Nature 570, 358 (2019)).

5) Page 6: In the discussion of the second-layer growth, it should be clearly stated if the second layer grows above or beneath the first graphene layer. As far as I could see, this question is never addressed in the manuscript, while the Supplement indicates nucleation of the second layer beneath the first. This is a key point, since the nucleation of the second layer, forced by the increased supersaturation, appears to occur near the edge of the first and also without any fixed lattice registry with the first layer. This could give important hints for future approaches to control the twist angle, which the authors may want to mention in the Discussion. Related to this point, I would caution against using the terms "Stranski Krastanov" and "Volmer Weber" growth in the

Supplement, since these have other meanings than those implied by the authors.

6) Page 8: The authors briefly mention the formation of a quasicrystal at 30 degrees twist, but again without citing the Science paper by Ahn et al., which demonstrated this Stampfli tiling for the first time, as well as Sutter et al., which showed the quasicrystal for metal dichalcogenides (see Point (2), above). The prior work needs to be cited here, and the discussion of the quasicrystal tiling should be expanded slightly to better explain it to the reader.

7) Methods: Replace "under 80 kV" and "under 2 kV" by "at 80 keV electron energy" etc.

8) Figure 3 (d): I strongly suggest adding a second histogram, magnified to cover only the region below 5 degrees twist angle, with much finer binning. In this way, the authors can document clearly which twist angles are difficult to obtain (near 0 degrees), due to the tendency of the second layer to snap into registry with the first. Also, there is an apparent contradiction between the histograms in Fig. 3 (d) and Fig. 4 (c). In one case, a lot of 0 degree stacking is found, while the other figure claims almost no 0 degree stacked flakes. Please clarify!

9) The device fabrication still involves a significant amount of impurities and bubbling, as shown in Supplementary Fig. 14, which in a way runs counter to the idea of a clean bottom-up synthesis. The authors may want to include a brief discussion/outlook, stating how these issues could eventually be overcome in an all-synthesis approach to making encapsulated heterostructures.

Response to referees

Response to the Reviewer #1

The manuscript reports an epitaxial growth method to grow tBLG on Cu foils. They can increase the ratio of tBLG to 86% (of total BLG?). They also used isotope to reveal the growth mechanism, and they attribute the formation of tBLG to the "turbulence" when changing gas-flow during the growth. The primary growth method has been reported in ref. 27 and 28. What different is the "turbulence". The method itself is not novel enough. Some of the evidences in this manuscript are not solid. My comments are as follows:

Response:

We sincerely thank the reviewer for the great effort in reviewing our manuscript. In this manuscript, the hetero-site nucleation strategy (*ex situ* nucleation strategy in original version) is the **key point** for increasing the fraction of tBLGs to all bilayer domains (86%, which is based on OM results; 88%, which is based on the SEAD results). This strategy that enables the nucleation of second layer graphene at different site from the original one of the first layer would alter the chemical environment around the nucleation site of second layer and therefore create the interlayer twist.

“Turbulence (now is replaced by “perturbation” according to the reviewer’s comments)” that is responsible for the hetero-site nucleation of the second layer is introduced by increasing both carbon source supply and hydrogen. According to the theoretical calculation, the termination of graphene edge could be modified by the partial pressure of hydrogen, which influences the growth of the second layer graphene (calculation result please refer to *J. Am. Chem. Soc.* **2014**, *136*, 3040). In our manuscript, we reported detailed experimental conditions for growing second-layer-free graphene or bilayer graphene, and we can successfully use different experimental conditions (ratio of carbon source supply and hydrogen) to realize the nucleation of a second-layer free domain, and to subsequently initiate the nucleation of the second layer according to the hetero-site nucleation strategy. In contrast, the reported method in ref. 27 and 28 (reported by our group) is based on the increase of carbon source supply for fast growth of monolayer graphene, which is responsible for new nucleation of monolayer graphene. Therefore, we believe that the hetero-site nucleation growth method and related detailed growth condition are important for growing tBLGs, and this method is novel enough for potential readers in

Nature Communications, and then we will address the reviewer's comments point by point in the following.

Question 1:

In the abstract lines 20-21, the author wrote: "Twisted bilayer graphene (tBLG) has recently attracted growing interest due to its unique twist-angle-dependent electronic properties. Fundamental investigations and applications of tBLG rely heavily on the availability of large-area tBLG with a broad range of twist angles." In the whole manuscript, there is no result related to "twist-angle-dependent electronic properties"; There is no demonstration that this method can benefit the "Fundamental investigations and applications". It needs more to show the potential, for example, how this growth method can benefit the field of twistrionics?

Response:

Thanks for the valuable comment and suggestion. We totally understand the reviewer's concern on the "twist-angle-dependent electronic properties" of as-obtained tBLGs. Indeed, the band structure of the graphene layers would be modified by the unique interlayer interaction in bilayer graphene. Specifically, the density of states (DOS) of tBLG near the Dirac points would be altered, leading to presence of van Hove singularities (vHSs), the positions of which are **twist-angle-dependent**. In detail, the energy difference between vHSs (ΔE_{vHS}) in conduction and valance band is calculated to be linear with $\sin(\theta/2)$ (*Phys. Rev. Lett.* **2012**, 109, 196802). Thus, the twist angle endows tBLGs twist-angle-dependent band structures, which contribute to the twist-angle-dependent electronic and optical properties. Following the reviewer's suggestion, we investigate the electronic structures of as-obtained tBLGs using the angle-resolved photoemission electron spectroscopy with spatial resolution better than 1 micron (micro-ARPES), and observed the twist-angle-dependent band structures.

Figure R1 shows the micro-ARPES data collected from three obtained CVD-grown tBLG domains on Cu substrate. By overlapping the Brillouin zones of graphene layers with ARPES intensity maps, the twist angle θ of $\sim 3^\circ$, $\sim 6^\circ$, and $\sim 11^\circ (\pm 1^\circ)$, respectively can be determined (Figures R1a-c). Here we also plotted the schematic diagrams of band structure for these tBLGs near the Dirac cones. The vHSs induced from interlayer coupling in band-crossing area are clearly observed in the energy-momentum-dispersion (the left panel of Figures R1d-f), where the top layer displays a higher

intensity in each figure. The corresponding integrated energy distribution curves (EDCs) are also plotted (right panel of Figures R1d-f) to quantitatively demonstrate the position of vHSs. Considering the substrate doping effect on the tBLGs, the ΔE_{vHS} can be calculated by the formula $\Delta E_{\text{vHS}} = 2|E_{\text{D}} - E_{\text{vHS}}|$, where E_{D} is the energy of Dirac point. Clearly, as the θ increases, the ΔE_{vHS} increases as well, consistent with previous reports (*Phys. Rev. Lett.* **2012**, 109, 196802). Our observation confirms that the electronic structure of our tBLGs is twist-angle-dependent, which endows tBLGs the θ -dependent-enhanced optical absorption, θ -dependent-enhanced Raman G-band intensity and enhanced photocurrent generation at certain wavelengths determined by θ (*Nano Lett.* **2014**, 14, 3353; *Phys. Rev. Lett.* **2012**, 108, 246103; *Nat. Commun.* **2016**, 7, 10699).

Figure R1. Electronic properties of tBLGs with various twist angles. (a-c) ARPES equal energy contours for the tBLGs with the twist angle of $\sim 3^\circ$ (a), $\sim 6^\circ$ (b), and $\sim 11^\circ$ (c), respectively. Inset: schematic diagrams of the reciprocal space BZs and two sets of Dirac cones of tBLGs. (d-f) Energy-momentum-dispersion along $K - K_\theta$ (left panel) with the twist angle of $\sim 3^\circ$ (d), $\sim 6^\circ$ (e), and $\sim 11^\circ$ (f), respectively. The corresponding integrated energy density curves (EDC) are presented in the right panel, where the vHSs are indicated. The dashed blank lines indicate the energy of the Dirac point (E_{D}).

Based on above results, we believe the as-grown tBLGs with twist-angle-dependent electronic band structure and properties can offer an ideal platform for the fundamental investigations and applications of tBLGs, such as the graphene-based photonics (*Nat. Commun.* **2016**, 7, 10699; *Phys. Rev. Lett.* **2020**, 124, 077401). According to the reviewer's suggestion, we have included the new ARPES results into the new version of our manuscript (Supplementary Fig. 23). We sincerely hope that the reviewer would agree that we share our results as soon as possible with readers.

Question 2:

In the abstract lines 23-24, the author wrote: "...current fabrication methods are hindered by the small twisted region size or the large portions of non-twisted bilayer graphene...". The mainstream to produce precisely twist angle control MLG samples is to use the "dry" transfer method to pick up part of exfoliated graphene then twist stack it on the remnant graphene. Please address where the "large portions of non-twisted bilayer graphene" comes from.

Response:

We very much appreciate the reviewer's comments and we apologize for incurred misunderstanding. During the past few years, many attempts have been made to experimentally obtain the tBLGs including SiC-epitaxial growth (*Nat. Phys.* **2009**, 6, 109), artificial-stacking approaches (*Nature* 2018, 556, 80; *ACS Nano* 2016, 10, 6725), and direct CVD growth methods (*Nat. Nanotechnol.* **2016**, 11, 426; *ACS Nano* **2013**, 7, 2587). As for tBLGs grown on SiC, study and applications of tBLG is strongly limited by the tedious transfer and the high price of SiC single crystal. Artificially stacked tBLG currently suffers from unavoidable interlayer contamination, which would affect interlayer coupling and also introduce an additional small rotation of graphene layer in some cases after the final stacking or during the annealing step. Consequently, although it is versatile for proof-of-concept demonstrations and fundamental studies, manual restacking process is clearly not scalable for practical technologies.

Currently reported CVD approaches for growing CVD-derived bilayer graphene still consists a large portion of AB-stacked (non-twisted) counterpart, owing to the higher stability of AB-stacked configuration. Thus, *"large portions of non-twisted bilayer graphene"* in CVD approach remains unsolved.

Thanks to the reviewer's comment, we revised the corresponding description in the abstract to avoid further misunderstanding as follow:

"The preparation of high-quality large-area bilayer with rich rotation angles would be important for the investigation of angle-dependent physics and applications, which, however is still challenging."

Question 3:

In the abstract line 27-29, "The portion of tBLGs with twist angles ranging from 0° to 30° were found to be improved to 86%, which is significantly higher than those reported previously." According to the data, I think 86% means 86% of total BLG counts are tBLG. Please make it clear in whole manuscript.

Response:

We very much appreciate the reviewer's comments and we apologize for our ambiguous expressions. 86% means 86% of total BLGs. Accordingly, we revised the clarified corresponding expressions of "the portion of tBLGs in BLGs" across the entire manuscript. In addition, the fraction of tBLGs to all bilayer domain ("86%"), mentioned in the main text, is currently replaced by 88%, which is based on the SEAD results to avoid misunderstanding.

Question 4:

In the abstract line 32-33, "Our study opens a new avenue for the controllable growth of tBLGs for both fundamental research and practical applications." According to the data in the manuscript, I do not think it is a "controllable growth of tBLGs", neither to achieve a precisely twist angle control or a large scale.

Response:

Thanks for the valuable comment. In this manuscript, the hetero-site nucleation strategy enabled the formation of tBLGs with enhanced portion compared to that of AB stacked one. Indeed, "to achieve a precisely twist angle control or a large scale" is final goal of controlled growth of tBLGs. Here, we strongly believe our results bring us one step closer to this goal, by overcoming the energy preference

of AB staking counterpart to tBLGs, with which the reviewer 2 also agree by stating that “85% is a new achievement for CVD growths”

For instance, by combining the reported approaches for controlling the nucleation density (*Nat. Nanotechnol.* **2016**, 11, 426) (grow mm-or cm-sized graphene single crystal) and our hetero-site nucleation method, it is possible to achieve the tBLGs with large domain size in the future, consistent with the reviewer’s suggestion “achieve a large scale”. This direction will continue to be explored in our future work. In addition, our approach is expected to be applicable to other two-dimensional materials with interlayer twist, based on the intentional design of the nucleation sites. Furthermore, to realize the precise control of twist angle, two approaches may be possible including by utilizing the axial screw dislocation (*Nature* **2019**, 570, 358; *Nature* **2019**, 570, 354) and by introducing intermediate layers (*Nat. Commun.* **2019**, 10, 5528; *Science* **2018**, 361, 782), which is also suggested by reviewer#3. Therefore, to address the reviewer’s concern, we eased revised the related description at the end of abstract, and made some revisions in the introduction part and discussion part:

At the end of the introduction section, sentence “*This new synthesis strategy by the intentional design of nucleation sites would bring more inspiration for the controlled growth of graphene and other 2D materials with precise control over interlayer twist in the near future*” was added.

In the discussion part, we revised our manuscript as follow:

This work describes a hetero-site nucleation strategy for successfully growing the tBLG with the fraction of tBLG to BLG higher than 88%, and a wide range of twist angle from 0° to 30°. By employing the gas-flow perturbation after the nucleation of first layer, the second layer nucleation could be initiated at a different location from that of the first layer, which was carefully investigated by isotope labeling in conjunction with micro-Raman spectroscopy. The sudden increase of carbon source supply would drive the system out of equilibrium, and allow the formation of the second-layer nuclei without the preferred equilibrium stacking order. In this work, since the second layer graphene is below the first layer, substrate plays crucial roles in determining the crystal orientation of graphene; the different local chemical environments of the nucleation sites of the two graphene layers are mainly responsible for the interlayer twist. The high quality of the as-grown tBLG was corroborated by clear Moiré patterns in the HR-TEM and carrier mobility exceeding $68,000 \text{ cm}^2 \text{ V}^{-1} \text{ s}^{-1}$ at room-temperature.

*There are some issues still need to be addressed in the future works. Firstly, although by intentional design of nucleation sites, the hetero-site nucleation strategy enables the formation of small twist angles ($\sim 3^\circ$). The torque toward smaller twist angles increases rapidly near the zero-angle, which should be overcome to obtain the very small twist angle (such as magic angle). Our work brings more inspiration for controlled growth of graphene and other 2D materials with interlayer twist, and future works still need be done to grow tBLGs with certain twist angles, possibly by controlling the nucleation density, designing the growth substrate (Adv. Mater. **2020**, 32, 2002034), utilizing the axial screw dislocation (Nature **2019**, 570, 358; Nature **2019**, 570, 354) or introducing intermediate layers (Nat. Commun. **2019**, 10, 5528; Science **2018**, 361, 782).*

Question 5-1:

Line 75, "...by introducing a gas-flow turbulence...". The author should explain the mechanism of the "turbulence". By changing the gas-flow, both pressure, temperature, gas concentration, and the ratio will change. If there is no information about what is truly in the secondary nucleation, other researcher can't repeat the experiment. Because different CVD systems will have a different process of "turbulence".

Response:

We very much appreciate the reviewer's comments and suggestions. Combining the valuable comments and suggestions of reviewer 3# (Question 3), we realized that the word "turbulence" is misleading to some extent, since it is a concept in hydrodynamics. Here we want to emphasize that the sudden increasing of carbon source supply (flow rate of CH₄) and partial pressure of hydrogen is essentially responsible for hetero-site nucleation and formation of tBLGs. To avoid misunderstanding, we replace the word "turbulence" by "gas-flow perturbation".

The increasing of partial pressure of hydrogen promote the formation of the hydrogen-terminated edges of the top layer graphene (Chem. Sci. **2014**, 5, 4639), which are beneficial for growing bilayer graphene owing to the two following reason: 1) Impeding the direct attachment of active carbon species onto the first layer; 2) Decoupling the interaction between the first-layer graphene and Cu substrate, which allow the diffusion of the carbon species beneath the first layer to fuel the growth of the second layer (J. Am. Chem. Soc. **2014**, 136, 3040) (Figure R2).

The sudden increasing of carbon source supply would produce an over-saturated supply of active carbon species, which would enable the nucleation of second-layer graphene with twist angle.

Figure R2. The influence of sudden increasing partial pressure of hydrogen and carbon supply on the secondary nucleation beneath the first layer.

In the following part, we would provide detailed discussion on the reason of the second-layer nucleating process.

Here, we'd like further explain our ^{13}C labeled growth experiments. In the ^{13}C labeled growth experiments shown in Figure 2b (please also see Figure R3), we can measure the grain sizes of each layer of tBLG at different times (5, 10, 15 and 20 min), and obtain the time evolution of grain size of the two graphene layers in a single domain, by analyzing the dosing sequence and 2D-band intensity Raman maps of ^{13}C and ^{12}C (Figure R3c,d). Our results imply that the grain sizes increase linearly with the growth time (Figure R3e), and therefore either top layer or second layer grow at a constant rate. By extending the fitting line (dash line in Figure R3e) to make it intersect with the x axis (time axis), the obtained intercept is corresponding to the time when the nucleation of 2nd layer graphene occurred, and the nucleation time is consistent with the time when we increased the carbon source and partial pressure of H_2 . For instance, in Figure R3, we increased the carbon source and partial pressure of H_2 at 10 min, consistent with the the nucleation time of 2nd layer graphene (10 min) (Figure R3e). Above results confirm the contribution of gas-flow perturbation to the nucleation of 2nd layer graphene. The nucleation sites can be spatially inferred by the Raman maps, which also shows the second-layer nucleation just occurred at the edge of ^{12}C shell of first layer at 10 min. To provide a vivid demonstration, we drew a series of cartoon figures to reflect the time evolution of the grain growth of tBLG (Figure R3f). In addition, we can also control the second-layer nucleation at 5 min by introducing the gas-flow perturbation at 5 min, and the corresponding data are shown in Figure 2a-c and g of our manuscript.

To further confirm that the nucleation of 2nd layer graphene is governed by the gas-flow perturbation, we introduced more cycles of gas-flow perturbation (sudden increase of CH₄ and H₂). In detail, we kept the flow rate of ¹²CH₄ as a constant value of 0.5 sccm, and introduced pulses of H₂ and ¹³CH₄ flow (2 sccm), at $t = 1, 2, 3, 4, 8, 13, 18, 20,$ and 22 min, and every pulse was kept for 10 s; the flow rate of H₂ was suddenly increased from 300 sccm to 1000 sccm at these timepoints and switched back to 300 sccm after 10 s. Because the amount of the H₂ and CH₄ in every pulse is higher than the amount that we normally used in hetero-site nucleation, the second layer graphene densely nucleate along the edge of first layer graphene (Figure R4a). Note that, the second layer graphene domains are difficult to continuously grow larger because the concentration of active carbon species would be quickly reduced into the original level after each pulse. By analyzing the zoom-in OM image (Figure R4b) and the dosing sequence (Figure R4c), we find that the second layer graphene just nucleated at the timepoints when we introduced these pulses. Furthermore, the carbon isotope distribution (Figure R4d,e) clearly show that the second layer graphene domains predominately consist of ¹³C, which further concludes that the introduced pulses lead to the formation of second layer graphene nucleation.

Figure R3. In depth explanation on the ¹³C labelling experiment corresponding to Figure 3b in the main text. (a) Feedstock feeding process for growth of isotope-labeled tBLG using the hetero-site nucleation strategy. ¹²CH₄ and ¹³CH₄ are alternately introduced with a duration of 5 min, and the

sudden increase of H_2 and CH_4 are proceeded at 10 min. (b-d) OM image (b), Raman 2D band intensity map corresponding to ^{13}C (c, $2D^{13}$ intensity map) and ^{12}C (d, $2D^{12}$ intensity map) of as-grown tBLG. (e) The grain sizes of 1st layer and 2nd layer graphene as functions of time, which indicate the nucleation time of both 1st and 2nd layer graphene (intersection of the curve and the horizontal axis). (f) Schematic diagram of the time evolution of tBLG growth, which is inferred from the Raman maps (c,d).

Figure R4. Excess perturbation results in multi-site nucleation. (a) OM image of a typical graphene domain, where there are a few laps of the second layer graphene. (b) A zoom-in image corresponding to the dashed box in (a). (c) Feedstock feeding process of H_2 , $^{13}CH_4$ and $^{12}CH_4$ for growing the graphene with multi-site nucleation of the second layer, during which H_2 and $^{13}CH_4$ pulses were introduced at timepoints of $t = 1, 2, 3, 4, 8, 13, 18, 20,$ and 22 min. The width of as-introduced pulses is 10 s. (d,e) Raman maps of G band intensity corresponding to ^{13}C (d) and ^{12}C (e), which indicates that the secondary nucleation of the second layer just occur at the time we introduce these pulses.

In the next part, we would provide detailed discussion on the reason of forming twist angle.

Question 5-2:

Line 123-132, do the author has any evidence or citation of this hypothesis? I suggest to do some simulation to support both "turbulence" and the hypothesis.

Thanks for the reviewer's comment. It was observed in above discussion that when introduce the gas-flow perturbation, the second layer would nucleate along the edge of the first layer, rather than at the original nucleation site of first layer. Next, we will give more **theoretical and experimental supports for the hypothesis in In Line 123-132 of our original manuscript to explain the nucleation behavior of second layer graphene**. According to the reported literature (*Adv. Sci.* **2018**, 5, 1700961), three key factors are mainly related to the nucleation: 1) nucleation barrier, 2) concentration of active carbon species and 3) diffusion barrier of the decomposed active carbon species. Specifically, the nucleation rate can be written according to the following formula:

$$R_{\text{nu}} = v_0 N^2 \sqrt{\frac{2G^*}{\pi k t N^*}} \exp\left(-\frac{E_b}{kT}\right) \exp\left(-\frac{G^*}{kT}\right) \quad (\text{R1})$$

Where N is the carbon atom concentration, G^* is the nucleation barrier, E_b is the diffusion barrier.

When the concentration of active carbon species is highly enhanced by the introduced gas-flow perturbation, the moving carbon species around the edge of first layer have **three** routes to participate into the growth or nucleation of graphene:

(1) **Attachment to the edge of first layer**. Because the graphene edges are terminated by hydrogen under the high partial pressure of H_2 (*J. Am. Chem. Soc.* **2014**, 136, 3040), attachment barrier of carbon species is enhanced, which lead to a reduced growth rate of first layer graphene. Thus, a high partial pressure of H_2 is essential for growing second layer.

(2) **Diffusion to fuel the nucleation of second layer graphene at the original nucleation site**. In this case, the carbon species must overcome following energy barriers (Figure R5): (I) the **diffusion barrier-I** corresponding to the barrier need to be overcome when diffusing from the bare Cu surface to graphene-covered region. According to the theoretical calculation, only carbon atoms can diffuse under the first layer to fuel the growth of the second layer graphene beneath the first layer (*J. Am. Chem. Soc.* **2014**, 136, 3040); (II) the **diffusion barrier-II** that is needed to be overcome when carbon atoms move underneath the first layer; (III) nucleation barrier of second layer graphene at the original nucleation site.

Figure R5. Energy barriers that the carbon species must overcome to forming the second-layer nuclei near the center of the first-layer.

Figure R6 (*J. Am. Chem. Soc.* **2014**, 136, 3040) shows the calculated energy profile of the C atoms that diffuse across hydrogen-terminated graphene edge (left) or metal-passivated graphene covered region (right), to fuel the nucleation underneath the first layer. Clearly, additional **diffusion barrier I** of 0.86 eV must be overcome if the C atom moves across the hydrogen-terminated graphene edge. This energy barrier would reduce the concentration of carbon atom for the nucleation at the original site, and increase the concentration of C atoms near the first layer graphene edge, which in turn promote the nucleation near the graphene edge of first layer graphene.

Figure R6. Diffusion barrier from bare Cu surface to graphene-covered region (from *J. Am. Chem. Soc.* **2014**, 136, 3040). (a) Schematic diagram and corresponding energy profile. (b,c) Diffusion barrier of the C monomer diffusing through the hydrogen-terminated graphene edge (b) and through metal-passivated graphene edge (c). By communication with the author's research group, we confirmed there exist minor mistake on the value 0.60 eV in panel b in the original published paper, and we corrected it here.

(3) **Nucleation at the edge of first layer (in our case)**. As discussed above, the concentration of carbon atoms for nucleation near the edge of first layer would be enhanced owing to the presence of **diffusion barrier-I**, which in turn would promote the nucleation at the edge. In addition, to form the second-layer nuclei, an energy barrier must be still overcome (nucleation barrier), the value of which is determined by the substrate characters, such as steps, kinks, and particles. The high-energy sites (active site) of the substrate in presence of steps, kinks, and particles would easily capture the moving C atoms to form larger the carbon clusters. This explanation is also reported by the literature (*Nat. Commun.* **2013**, 4, 2096). Experimentally, we found that the surface roughness near the nucleation site of either the first or the second layer of graphene is higher than other regions (Figure R7). In addition, the

different chemical environments of these active site from the original nucleation site would result in the formation of a different orientation of second layer from the top layer.

Figure R7. One example of substrate effect on the hetero-site nucleation of the second layer graphene. (a) SEM image of a tBLG. (b) Schematic diagram displaying the nucleation site of the second layer. (c-e) AFM height images zoomed in the region near the nucleation center. (f-h) AFM peak force error images corresponding to (c-e), which show clearer edges of the tBLG.

Thus, based on the calculation results, we believe that the presence of **diffusion barrier-I** and termination of graphene edge by hydrogen in our case are the main reason for nucleation of second layer graphene at the edge of first layer and formation of interlayer twist. Accordingly, we revised our manuscript as follow:

*Generally, when the concentration of active carbon species is enhanced, the nucleation rate and the growth rate would be increased (Adv. Mater. **2016**, 28, 4671; Nano Res. **2016**, 10, 355; Adv. Sci. **2018**, 5, 1700961). However, because the partial pressure of H_2 is greatly enhanced simultaneously,*

the graphene edge would be terminated by hydrogen, which hinders the attachment of the active carbon species to the edge of the first layer (ACS Nano 2011, 5, 6069; Nano Res. 2016, 10, 355). According to the theoretical calculation, only carbon atoms can diffuse under the first layer to fuel the growth of the second layer graphene beneath the first layer (J. Am. Chem. Soc. 2014, 136, 3040; Nat. Nanotechnol. 2016, 11, 426). Thus, if the nucleation of second layer occurs at the same site with that of first layer, several energy barriers must be overcome (Supplementary Fig. 6, Figure R5): 1) diffusion barrier-I corresponding to energy barrier needed to be overcome when carbon atoms diffuse from the bare Cu surface to graphene-covered region, 2) diffusion barrier II needed to be overcome when carbon atoms diffuse beneath the first layer graphene, and 3) nucleation barrier of the second layer graphene. The diffusion barrier-I is reported to be ~0.6 eV according to previous work (J. Am. Chem. Soc. 2014, 136, 3040) so that it will impede the diffusion of carbon atom to the graphene-covering region, and increase the concentration of carbon atoms near the first layer graphene edge, which in turn promote the nucleation near the graphene edge. Near the edge of first layer, the high-energy sites (active site) of the substrate in presence of steps, kinks, and particles would capture the C atoms and result in the nucleation of second layer, consistent with the reported literature (Nat. Commun. 2013, 4, 2096). Therefore, when gas-flow perturbation is introduced, the second layer preferentially nucleates near the growth frontier of the first layer, rather than at the original nucleation center, and the local environments near the nucleation sites of the two graphene layers are therefore different, which is responsible for the formation of interlayer twist (Supplementary Fig. 7, Figure R6).

Question 6:

Line 80-82 and Fig. 3d. Twistronics is sensitive to twist angle. I do not think the edges of graphene domains can be used to precisely determine the twist angle. Fig. 1b clearly shows that none of the graphene domains is a perfect hexagon. Some of them even have clearly nonparallel opposite sides. How did the author determine 0 degree? If they cannot, where this "86%" comes from?

Response:

Thanks for the valuable comments. Indeed, the morphology of as-grown tBLGs are not so perfect hexagon, which is influenced by the crystallographic orientation (*Nano Lett.* 2013, 13, 5692), the etching of H₂ (*ACS Nano* 2011, 5, 6069), and growth rate (*Proc. Natl. Acad. Sci. U. S. A.* 2012, 109, 15136; *Nat. Nanotechnol.* 2016, 11, 930). We agree that it is not precise way to determine the twist angle by measuring the edges of graphene domains in OM image, in comparison with SAED pattern in TEM

characterization. However, using domain edge to estimate the twist angle is rapid and compatible with following device fabrication. Since the H-terminated zigzag edge is the most stable edge considering our growth parameters (*Chem. Sci.* **2014**, 5, 4639), the sharper edges of graphene islands are widely used to roughly determine the crystallographic orientation (*Adv. Mater.* **2015**, 27, 1376; *ACS Nano* **2017**, 11, 12337; *ACS Nano* **2018**, 12, 6117). Specifically, for the oblong-hexagonal islands, the long side is usually used to be defined as the zero-degree side (*Nano Lett.* **2014**, 14, 5706). Following the reviewer's suggestion, we systematically compared the results (twist angle) derived from edge direction of domains in OM image and corresponding TEM SAED results of various graphene domains to improve accuracy of OM-based method.

As the reviewer mentioned, the graphene domains are not perfect hexagon, and some of domain edges are even nonparallel to the opposite sides. This indeed results in the uncertainty in the measurement of the twist angles. To evaluate the error (inaccuracy error) of the twist angle obtained based on edge direction of domain, we did a series of experiments and established rules for measuring twist angle based on edge direction of domain to ensure the corresponding result is closer to the derived results from SEAD pattern. We summarized 4 types of typical tBLG morphologies that we usually obtained and the rules for measuring the twist angles (Table R1, Figure R8-R11).

Table R1. Rules for measuring the twist angles.

Type	Description	Rules	Schematic diagrams
I	Regular hexagon	Take any two corresponding edges of each layer (dashed blue and red line), and measure the included angle between two lines		Oblong-hexagon, internal angles of which are still all 120°		
II	 Oblong-hexagonal shape All opposite edges of the hexagon are parallel but the angle between the long side and the adjacent side is not 120° 	Take the longest sides as referring lines, and measure the included angle between two longest lines	III	 Oblong-hexagonal shape One or two pair of opposite edges of the hexagon are not parallel The long edge of the hexagon is parallel to the opposite edge 	Take the long sides as referring lines, and then measure the included angle between two longest lines	IV	 Oblong-hexagonal shape The long edge of the hexagon is not parallel to the opposite edge 	Take the long sides as referring lines, and then measure the included angle between two longest lines, but the error is larger than type III		 Oblong-hexagonal shape None of the pair of opposite sides are parallel 		

Type I: Regular hexagons of each layer of graphene (or Oblong-hexagonal tBLG of which all the internal angles are 120°). In this case, we take any two corresponding edges of each layer, and measure the included angle between two lines. Figure R8 shows a typical example. We transferred the tBLG onto a coordinated TEM grid, and took an OM image with enhanced contrast to visualize the edge of graphene domain, and therefore we got the twist angle of 12.9° (Figure R8a). Then we

characterized this tBLG domain using TEM image at low magnification which has a higher resolution than OM images (Figure R8b). Figure R8c shows how we measure the orientations of the edges of each layer and calculate the twist angle. We carried out the selective area electron diffraction (SAED) at different positions of this tBLG domain, which is shown in Figure R8d (monolayer region) and Figure R8e-h (tBLG regions). Note that, here we made a rotation of obtained SEAD pattern according to the magnetic rotation angle under different magnifications and diffraction mode. According to the Fourier transform relationship, we can conclude that the edges of the tBLG are indeed zigzag terminated as shown in Figure R8c, and the derived crystal orientations at different regions from SAED results are almost same. By measuring the included angle between the two SAED patterns of each layer, we can obtain the twist angle of 12.1° (Figure R8g-h). Consequently, the above rule is applicable to this type of tBLG: the twist angle measured by OM method is very similar to that obtained from SAED patterns.

Figure R8. Type I. (a,b) OM and TEM images of typical tBLG on coordinated TEM grid. (c) Schematic diagram of measuring the twist angle. (d) SAED pattern of monolayer taken from the region marked with blue circle in (b). (e-h) SAED patterns of tBLG taken from the regions marked with red circles in (b).

Type II: Oblong-hexagonal shape with parallel opposite sides but the angle between the long side and the adjacent side is not perfect 120° . As shown in Figure R9a,b, we take the longest edges of different layers as the referring edge to measure the twist angle of 11.0° . From the SAED pattern and Fourier transform relationship, we confirmed that the longest side is terminated by zigzag edge (Figure

R9c), and that both the first layer and the second layer graphene are single-crystalline (Figure R9d-h). Derived from SAED pattern, the twist angle is 11.0° , similar to the results based on the edge direction in OM-based method.

Figure R9. Type II. (a,b) OM image (a) and TEM image (b) of typical tBLG on coordinated TEM grid. (c) Schematic diagram of measuring the twist angle. (d,e) SAED pattern of monolayer taken from the region marked with blue circle in (b). (f-h) SAED patterns of tBLG taken from the regions marked with red circles in (b).

Type III: Oblong-hexagonal shape, one or two pair of opposite sides are not parallel, the long side is parallel to the opposite side. In this case, we can also choose the longest edge as the referring edge to get the twist angle of 28.5° (Figure R10a-c). From the SAED pattern and Fourier transform relationship, we confirmed that the longest side is terminated by zigzag edge (Figure R10c), and that both the first layer and the second layer graphene are single-crystalline (Figure R10d-h). Note that, if the included angle (θ_l) between two selected edges is larger than 30° , we can safely subtract this value from 60° to obtain the twist angle (Figure R10c). Derived from SAED pattern, the twist angle is 28.8° , similar to the results based on the edge direction in OM method.

Consequently, for all type I, II, and III, based on the measurement of edge direction, we can get a similar twist angle with the angle that we derived from SAED patterns.

Figure R10. Type III. (a,b) OM image (a) and TEM image (b) of typical tBLG on coordinated TEM grid. (c) Schematic diagram of measuring the twist angle. (d,e) SAED pattern of monolayer taken from the region marked with blue circle in (b). (f-h) SAED patterns of tBLG taken from the regions marked with red circles in (b).

Type IV: Oblong-hexagonal shape, none of the opposite sides are parallel or the long side is not parallel to the opposite sides. In this case, we also use the longest edge as the referring edge and get two possible twist angle of 4.0° and 7.0° (Figure R11a-c). This is because the opposite sides of the tBLG are not parallel, the longest edge are not perfect zigzag terminated (Figure R11c). Derived from SEAD pattern, the twist angle is 4.9° , corresponding to 4.0° based on the measurement of edge direction. Note that, both the first layer and the second layer graphene are single-crystalline, which is confirmed by SAED characterizations (Figure R11d-h)

Therefore, the twist angle of type-IV graphene domains would have a larger error, based on the measurement of domain edge direction.

Figure R11. Type IV. (a,b) OM image (a) and TEM image (b) of typical tBLG on coordinated TEM grid. (c) Schematic diagram of measuring the twist angle. (d) SAED pattern of monolayer taken from the region marked with blue circle in (b). (e-h) SAED patterns of tBLG taken from the regions marked with red circles in (b).

Based on above discussion, we should avoid measuring the twist angle using the domain edge direction for type IV tBLGs, and SEAD would be the best way to determine the twist angle. In addition, we believe growth of tBLGs on single-crystal Cu substrate (for instance, Cu(111)) would reduce the portion of type IV, because the formation of irregular shape is usually caused by substrate-mediated nonuniform diffusion of carbon species (*Nano Lett.* **2013**, 13, 5692; *Nano Res.* **2016**, 9, 2182).

To address the reviewer's concern and to avoid further misunderstandings, we replaced some typical images with regular hexagonal tBLG domains in Figure 1b. Additional discussion about the rules and error of measuring the twist angle using domain edge has been included in our new version (Supplementary Note 5, Supplementary Table 1, and Supplementary Fig. 9,10). Furthermore, the portion of tBLGs to all bilayer domain ("86%"), mentioned in the main text, is currently replaced by 88%, which is based on the SEAD results to avoid misunderstanding.

Figure R12. The revised Figure 1b.

Question 7:

Fig. 3c should be together with their statistical distributions. Now it is meaningless.

Response:

We very much appreciate the reviewer's comments and suggestions. The statistical distributions corresponding to the pie charts of figure 3c in our original manuscript are shown in Figure R13b. According to the comments and suggestions of reviewer 2# (Question 1 and Question 2) and reviewer 3# (Question 8), we realized that the statistical distributions in Figure 3c (OM-based) and Figure 4d (SAED-based) would cause misunderstanding. Therefore, we move Figure 3c into supporting information as Supplementary Fig. 13 and added the corresponding statistical distributions.

Figure R13 (Supplementary Fig. 13 in our revised Supplementary Information). Ratio of tBLG and AB-BLG and distribution of twist angles in as-grown tBLGs obtained by measuring the edges of hexagonal graphene domains using OM results. (a,b) Pie charts of stacking order (AB-BLG or tBLG) for graphene growth without hetero-site nucleation strategy (a) and the corresponding distribution of twist angles (b). (c,d) Pie charts of stacking order (AB-BLG or tBLG) for graphene growth with hetero-site nucleation strategy (a) and the corresponding distribution of twist angles (b).

Question 8:

Line 173-175, do the author have evidence or citations? Why in Fig. S11, there are some small twist angle samples?

Response:

Thanks for the valuable comment and suggestion. In line 173-175 of our manuscript, we show some HR-TEM images of as-grown tBLGs, where the graphene superlattices can be interpreted as clearly observed Moiré patterns. The diffractograms, which are obtained by using a fast Fourier transform (FFT), reflect the twist angles.

In previous work regarding growth of tBLGs, almost no small twist angle was observed. This is because, taking the substrate effects into consideration, if the two graphene layers share the same nucleation site, the same surrounding microscopic environment would result in the same orientation of both layers, which would reduce the portion of small twist angles. Therefore, it is very difficult to obtain tBLGs with small twist angles ($< 3^\circ$), consistent with the reported works (*Nano Lett.* **2012**, 12, 1609; *ACS Nano* **2013**, 7, 2587). Here, by employing the hetero-site nucleation strategy, the orientations of the two graphene layers are affected by different local environments with presence of kinks, particles and steps, which enables the formation of small twist angles. Thus, by using the hetero-site nucleation strategy, we indeed obtained some tBLGs with small twist angles such as $\sim 1.5^\circ$, $\sim 2.6^\circ$, $\sim 2.9^\circ$, $\sim 3.5^\circ$ (Figure 4 and Supplementary Fig. 16).

It should be noted that the portion of small twist angle is still limited (Figure 4c). In our opinion, the reason are as follows:

Regardless of the influence of the substrate, AB-stacking counterpart is the more stable configuration, and as the twist angle increases, the energy of the tBLGs gradually increases, and tBLGs become relatively unstable (*Physical Review B* **2011**, 84, 195421). It was reported previously that during growth, tBLG will be subjected to torque, which would drive it toward smaller twist angles, and eventually zero twist angle. The magnitude of the torque is proportional to the derivative of the interlayer potential and thus increases rapidly near the zero angle (*Nano Lett.* **2012**, 12, 1609). Thus, during the CVD growth process, especially at the nucleation stage, the orientation of the second layer graphene would be subjected to the torque toward smaller twist angles and zero angle to form stable AB-stacked configuration (Figure R14).

Figure R14. Potential energy and corresponding torque per atom in the tBLG system. (*Physical Review B* **2011**, 84, 195421; *Nano Lett.* **2012**, 12, 1609)

According to the reviewer's suggestion, we have included the above discussion into new version to explain the formation of small-angle tBLGs in our case (please see the caption of Supplementary Fig. 18).

Question 9:

Line 183-207, again, if there are no twist angles related to electrical properties, it is meaningless.

Response:

Thanks for the valuable comment and suggestion. Referring to the response of Question 1, we conducted the ARPES characterizations and imaged the twist-angle-dependent band structures of tBLG with various twist angles, which would result in the twist-angle-dependent electronic and optical properties. Furthermore, as we observed in HRTEM, tBLGs with various twist angle would produce Moiré patterns with tunable Moiré pattern period. The formation of Moiré period along with the presence of vHSs will surely influence the electronic properties of graphene (*Phys. Rev. B* **2014**, 90, 155451). The electronic transport measurement in Figure 5 has proven the high quality of as-grown tBLG since the room-temperature mobility exceeds $68,000 \text{ cm}^2 \text{ V}^{-1} \text{ s}^{-1}$, comparable to the manually stacking one. Thus, we believe based on the tBLGs grown on our method, it will be possible to investigate the twist-angle-dependent electrical properties of tBLGs. However, our main focus is

material synthesis, and we sincerely hope the reviewer could understand this. To address the reviewer's concern, we have included the ARPES data and above discussion into new version.

Response to the Reviewer #2

This submission to Nature Communications presents a novel technique for growing twisted bilayer graphene over a wide range of twist angles. Both comprehensive growth methods and characterization (primarily Raman, TEM and electrical mobility) confirm the utility and promise of the new 'ex-situ' growth approach presented.

CVD grown twisted bilayer graphene is common, but the current methods favouring tBLG growth are haphazard at best. Numerous groups cited have already achieved CVD growth of bilayer graphene where >50% of bilayers twisted (e.g. Hone, Park, and many others, as confirmed by darkfield TEM) for nearly a decade. However, all CVD growth that I am aware seem largely random occurrences despite what the literature may claim. Typically, subtle changes in foil-selection or differing furnace result in entirely different growth patterns of the bilayer. Most often, the second layer is either all Bernal (AB stacked) or the angle distribution is poor. This work appears to break this 'trial-and-error' approach to CVD tBLG growth by developing a second layer induced nucleation trigger by induced turbulence of furnace airflow. This is best evidenced by comparing Fig. 3d and 4c that show the AB growth are reduced from 85 to 15%. I am aware of papers that claim up to 50% tBLG over AB stacked, but 85% is a new achievement for CVD growths (to my knowledge). This novel approach merits broad communication given the recent topical interest in tBLG (owing to its to unexpected superconducting and Mott insulator properties seen in exfoliated samples).

As someone who does not grow CVD graphene, I can only comment on the utility to the experimental community and the quality of characterization methods employed. I believe after minor revision both are sufficiently high to merit publication in Nature Communications. Nonetheless, reproducibility of the growth method proposed on alternate setups remains a concern. Ultimately I must

yield to CVD growth experts to better access the novelty and reproducibility of this triggered second-layer growth mechanism.

The quality of data presentation and range of twist angles is achieved is a sufficiently exciting advance to me. I remain confident that most of the novel many-body electronic properties observed in exfoliated graphene will be reproduced in CVD samples such as the growths outlined in this paper. I hope this work will accelerate such discoveries.

Response:

We appreciate very much for the positive comments from the reviewer on the quality of our work and the explicit recommendation of publication. After carefully reading the reviewer's comments, we have summarized the reported fraction of tBLG to all obtained bilayer domains in the previous literatures. Here, we listed these reported data and revised the corresponding discussion in our manuscript:

“Thus, compared to AB-BLG, the fraction of tBLG were limited to around 50% in previous reports”.

Table R2. Reported data of fraction of tBLGs in BLGs

Group	Fraction of tBLGs	References
Xiangfeng Duan	10%	ACS Nano 2012 , 6, 8241-8249
Jiwoong Park	30%	Nano Lett. 2012 , 12, 1609
Po-Wen Chiu	50%	ACS Nano 2013 , 7, 2587-2594
James M. Tour	52%	Angew. Chem. Int. Ed. 2014 , 73, 1565-1569
Shigeo Maruyama	~0%	ACS Nano 2014 , 8, 11631-11638
Judy Wu	10%	Carbon 2015 , 93, 199-206
Weiwei Cai	17%	Small 2015 , 11, 1418-1422
Gui Yu	10%	J. Mater. Chem C 2016 , 11, 1418-1422
Hiroki Ago	20-30%	Chem. Mater. 2016 , 28, 4583-4592

Hengxing Ji

7%

Chem. Mater. **2018**, 30, 7852-7859

Hailin Peng

50.3%

ACS Nano **2020**, 14, 2 1656-1664

We will address the reviewer's comments point by point as follows.

Question 1:

My primary concern remains that this paper does not characterize the twist angles with the accuracy needed for the broad community intended. Throughout they only characterize tBLG with an approximate angle (e.g. ~6 degrees). However, both TEM and optical methods can achieve far more precise angle characterizations, often to the first decimal point.

What is the error bar on the angle characterization? Can graphs such as Fig. 3d and 4c be binned smaller than 2-degree increments? This would be especially helpful in the 0-2 degree bin.

To attract broad interest this paper needs more convincing data that <3 degree tBLG growths are likely by this method. Otherwise, the intended audience will not be convinced that 'magic angle' flatband electronic behaviour is promising from this CVD growth method.

More careful aperture and apertureless TEM (esp. for some lower angle cases) is strongly recommended.

Response:

We understand the reviewer's concern on the error of determining the twisted angle by TEM or optical methods, and we appreciate very much for the valuable suggestion. The effective bin width in our histogram distribution should be indeed determined by the error of the methods for characterizing the twist angle. Since the TEM method is a more precise way to characterize the twist angle than the OM approach, the error bar of TEM (SAED) method should be smaller than that of optical method. To address the reviewer's concern, we analyze the error (inaccuracy error) of twist angle measured in two cases: 1) OM image of tBLG on SiO₂ (Figure R15), and 2) SAED patterns of the tBLG (Figure R16).

1) **Error analysis of OM-method.** Figure R15a shows a typical OM image ($\times 100$ objective) of a tBLG transferred on SiO_2 , and the twist angle is obtained by measuring the included angle between the edge of hexagonal domains of each layer. However, error is inevitable when measuring the orientations of each edge, because the limited resolution of OM image would contribute to the deviation of the edge recognition. As shown in Figure R15b, the parallel white dashed lines indicate the real direction of the edge. Usually, the optical resolution σ is around $0.2 \mu\text{m}$, which responds to the size of fuzzy region. Therefore, the error in determine one edge orientation is

$$\text{Error}_\varphi = \arctan \frac{\sigma}{L} \approx 0.9^\circ \quad (\text{R2})$$

Where L is the length of domain edge, which usually range from ten micrometer to tens of micrometers. In this case, the Error_φ is calculated to be around 0.9° . Since the twist angle (θ) is determined by measuring the included angle between two edges of the two layers, the OM-based error of twist angle (Error_θ) should be twice the value of Error_φ :

$$\text{Error}_\theta = 2 \times \text{Error}_\varphi \approx 1.8^\circ \quad (\text{R3})$$

Then, we carefully measured the orientations of each pair of hexagonal edges and calculated the corresponding included angle (twist angles) (θ_i , $i=1\sim 6$) as shown in Figure R15c. The range of θ_i is 3° , and the standard deviation is about 1.2° , slightly smaller than Error_θ . The deviation of measured θ_i is caused not only by the error of OM-based method, but also by the asymmetric graphene growth, the anisotropic thermal expansion during the cooling process, and the formation of the Cu steps (*Small* **2018**, 14, e1800725; *Phys. Rev. Lett.* **2018**, 120). In addition, Raman spectroscopy characterization under 532 nm laser ($E_{\text{laser}} = 2.33 \text{ eV}$) was conducted to further determine the twist angle shown in Figure 15a. The G-band intensity in tBLG region display a 48-fold enhancement as compared to that of monolayer region (Figure R15d), indicating a presence of twist angle of $\sim 12^\circ$ according to the formula $\theta = 3aE_{\text{laser}}/4\pi\hbar v_f$ (the E_{laser} matches very well with the van Hove singularity in tBLGs) (*Phys. Rev. Lett.* **2012**, 108, 246103). Figure R15e shows the Raman G-band intensity maps, where a high uniformity indicates single crystal nature of tBLG without spatial change in twist angle, since the G-band intensity is very sensitive to the twist angle (*Nano Lett.* **2012**, 12, 3162).

Figure R15. Error analysis on the twist-angle measurement of tBLG on SiO₂ by OM method. (a) A typical OM image of tBLG on SiO₂. (b) Illustration of the origin of the one-side direction error during the measurement. (c) Twist angles measured between 6 pairs of hexagonal edges of tBLG. (d,e) G-band intensity map (d) and typical single point Raman spectra of ~12°-tBLG and monolayer graphene (e).

2) Error analysis of TEM (SAED) method. The SAED characterization is a widely used method, which is capable to determine the crystal orientation with a high credibility (*ACS Nano* **2018**, 12, 6117). Since the diffraction points turn fuzzy when zoomed in, there still exists an inevitable error. Firstly, we analyzed the SAED results which were used in the statistics in our manuscript. Taking the ~2.6° SAED pattern as an example (Figure R16, see also in Supplementary Fig. 17 in our revised supplementary information), the weight dashed lines we plotted along opposite diffraction points intersect each other with a twist angle. The error of determining the orientation of each SAED pattern (Error_φ) can be estimated by the full width at half maximum (FWHM) of the fuzzy diffraction points and the distance between each opposite bright points in single SAED pattern (bottom panel, Figure R16a).

$$\text{Error}_{\varphi i} = \arctan \frac{FWHM_i}{2K_i} \quad (\text{R4})$$

where $i=1,2,3\dots$, indicating the diffraction order, K_i is the distance from the center of the Brillouin zone to the diffraction point **with the diffraction order of i** . To further estimate the error of TEM-based method using different order of diffraction points, we plot the line profiles near the 1st, 2nd, 3rd, and 5th diffraction points (Figure R16b-e), where the FWHMs are all about 0.8 nm^{-1} . The vector of diffraction point K_i increases with the diffraction order i ($K_1= 4.69 \text{ nm}^{-1}$, $K_2= 8.12 \text{ nm}^{-1}$, $K_3= 9.37 \text{ nm}^{-1}$, $K_5= 14.06 \text{ nm}^{-1}$), indicating that the value **Error $_{\phi}$** would be smaller if we choose higher order of diffraction points. Similar with the OM method, since we need measure the orientations of each layer, the error of twist angle (**Error $_{\theta}$**) should be two folds of the **Error $_{\phi}$** , and the obtained **Error $_{\theta}$** using the points of different diffraction orders are shown in Table R3.

Table R3. Errors of SAED method by measuring the tilt angles of diffraction points with different diffraction orders

Selected diffraction order (i)	1	2	3	5
$K_i \text{ (nm}^{-1}\text{)}$	4.69	8.12	9.37	14.06
Tilt angle error: Error $_{\phi}$ (°)	0.49	0.28	0.24	0.16
Twist angle error: Error $_{\theta}$ (°)	0.98	0.56	0.48	0.32

Figure R16. Error analysis on the twist-angle measurement of the tBLG domain by TEM (SAED) method. (a) A typical SAED pattern (top) and the illustration on the error of the tilt angle determined by one sets of diffraction points. (b-e) Diffraction intensity profiles versus position of four diffraction points shown in (a), where FWHMs are marked.

Next, to address the reviewer's concern, we collect the SEAD patterns using different the aperture and camera length. The SAED results (Figure R17 and Figure R18) are collected by using the aperture of 200 nm and 800 nm, respectively. The camera length (L) is another important parameter related to the error, since the distance from the diffraction point to the diffraction center on the screen (R) is proportional to L : $R = L\lambda/d$, where λ is the wavelength of the electron beam, and d is the lattice of graphene. Therefore, a larger the camera length would allow us obtain a larger R on screen and therefore a smaller error. To confirm this, we collected SEAD patterns by using the camera length of 750 mm to 340 mm. The FWHM of the diffraction points increases as the camera length decreases. Consequently, Error_θ measured by using the 1st and 2nd order diffraction points can reach 0.44° with camera length 750 mm or 560 mm (Table R4).

Figure R17. SAED conducted by using aperture of 200 nm. (a-c) SAED pattern conducted with camera length of 750 mm. (d-f) SAED pattern conducted with camera length of 560 mm. (g-i) SAED pattern conducted with camera length of 340 mm.

Figure R18. SAED conducted by using aperture of 800 nm. (a-c) SAED pattern conducted with camera length of 750 mm. (d-f) SAED pattern conducted with camera length of 560 mm. (g-i) SAED pattern conducted with camera length of 340 mm.

Table R4 (Supplementary Table 3). Errors of SAED method by measuring the tilt angles of diffraction points with different diffraction orders

Aperture (d, nm)	Camera length (mm)	diffraction order (i)	K_i (nm ⁻¹)	Tilt angle error: Error _φ (°)	Twist angle error: Error _θ (°)
200	750	1	4.69	0.22	0.44
		2	8.12	0.22	0.44
	560	1	4.69	0.34	0.68

	340	2	8.12	0.22	0.44
		1	4.69	0.32	0.64
		2	8.12	0.27	0.54
800	750	1	4.69	0.23	0.46
		2	8.12	0.22	0.44
	560	1	4.69	0.30	0.60
		2	8.12	0.24	0.48
	340	1	4.69	0.31	0.62
		2	8.12	0.27	0.54

In summary, the above analysis indicates that the error bar of OM characterization should be 1~2°, and error bar of TEM (SAED) characterization can be lower than 0.5°. Hence, we are afraid we couldn't make the bin width of Fig. 2d (in our original manuscript, OM-characterized histogram) smaller than 2°. To address reviewer' confusion, we revised the histogram distribution obtained from TEM characterization (**bin width is set to 1°**) at the main text, and put the statistical data obtained from OM characterization to supporting information (Supplementary Fig. 13 (Figure R19)), and included the above discussion to avoid further misunderstanding (Supplementary Fig.9, 10, and 17; Supplementary Table 1 -3). In addition, to address the reviewer's concern about the distribution below 3°, and combining the **reviewer #3's suggestion**, we added a second histogram into the new version of supporting information (Supplementary Fig. 18 (Figure R20)), showing the region below 5 degrees twist angle, with the bin width of 0.5° based on SAED results.

Figure R19 (Supplementary Fig. 13). Ratio of tBLG and AB-BLG and distribution of twist angles in as-grown tBLGs obtained by measuring the edges of hexagonal graphene domains using OM results. (a,b) Pie charts of stacking order (AB-BLG or tBLG) for graphene growth without hetero-site nucleation strategy (a) and the corresponding distribution of twist angles (b). (c,d) Pie charts of stacking order (AB-BLG or tBLG) for graphene growth with hetero-site nucleation strategy (a) and the corresponding distribution of twist angles (b).

Figure R20 (Figure 4c and Supplementary Fig. 18). Statistical results of twist angles. (a) Statistical results of stacking order (AB stacking or non-AB stacking) and distribution of twist angles based on SAED patterns of as-grown BLGs. (b) A second histogram which is magnified to cover only the region below 5° twist angle, with the bin width of 0.5° . Here, the hetero-site nucleation strategy, in which

different local environments plays crucial roles in determining the orientations of two graphene layers, enables the formation of small twist angles ($< 3^\circ$).

Question 2:

Comparing Fig. 3d and 4c, I may have missed a comparison of what % of the overall growth areas were single layer vs. bilayer. Systematic minimization AB growth is important, but only if it does not substantially reduce the overall prevalence of bilayer vs. single layer coverage.

Response:

We thank for the reviewer's valuable comments. The hetero-site nucleation method enables the formation of the tBLGs, because the orientations of the two layers are determined separately by different local chemical environment. The overall coverage of bilayer graphene can reach over 85%. Geometrically, the ratio of the second layer graphene is influenced by 1) the distance between the nucleation sites of the first layer graphene and second layer graphene (\mathbf{d}_1) and 2) the distance between the edges of first layer graphene and second layer graphene (\mathbf{d}_2) (Figure R21 a,d).

As shown in our manuscript, the hetero-site nucleation process is stimulated by the sudden increasement of H_2 and CH_4 , and the nucleation time and nucleation site of the second layer graphene (relative to those of the first layer) is proved to be controllable. Hence, \mathbf{d}_1 can be roughly tuned by changing the time points when we increase the H_2 and CH_4 . In another hand, higher partial pressure of H_2 would be effective to decreases the growth rate of the first layer graphene, and the resulted hydrogen-terminated edges would allow more carbon atoms diffuse beneath the first layer and fuel the growth of second-layer graphene. Thus, higher partial pressure of H_2 would help decrease the value of \mathbf{d}_2 . Therefore, by tuning the time when we introduce gas-flow perturbation to initiate the nucleation of the second layer nucleation and the partial pressure of H_2 , the overall coverage of bilayer graphene can be optimized to reach over 85% (Figure R21).

Figure R21. Regulation of the area ratio of bilayer graphene versus the overall area.

Question 3:

Searching for information on intrinsic doping and/or Fermi level characterization of the growth came-up absent. As Hall-bar conductance is presented in Fig. 5b, why not estimate the Fermi-level of the sample shown and comment more on the tBLG doping level generally. Hysteresis in RT conductance measurements (eliminated by cooling) can make this difficult, but rough estimates should be possible (and helpful)?

Response:

We understand the reviewer's concern on the doping level of obtained devices. Indeed, in Figure 5b, it is the typical **Raman spectra** of tBLG on SiO₂ and encapsulated in hBN respectively. Based on Raman results, we can extract the G-peak positions (ω_G), 2D peak positions (ω_{2D}) and full width at half-maximum (FWHM) of the 2D peak (Γ_{2D}). By comparing the relationship between ω_{2D} and ω_G in each sample (the colored scatter diagram, Figure 5c), we find that the corresponding data of h-BN encapsulated sample locates at the lower left corner with narrower 2D peaks, This observation indicates

less strain and doping level of h-BN encapsulated sample than those of tBLG on SiO₂ (*Nat. Commun.* **2012**, 3, 1024).

Thanks to the reviewer's kind suggestion, we extracted the doping level and Fermi-level of the sample from the multi-channel transport measurements. From transfer curves, we can extract the position of neutral points (Dirac points), and there exist a slight deviation from the zero point. Considering the parallel capacitance model, we calculated the concentrations of residual carriers and corresponding Fermi-levels as shown in Figure R22.

According to the reviewer's suggestion, we have included the results about the Fermi level of tBLGs into the new version.

Figure R22. Doping level and Fermi-level of tBLG in the Hall bar device. (a) The resistance as functions of gate voltage with the source-drain of 1-5. (b) OM image of the encapsulated tBLG Hall bar device, where the electrode numbers are marked. (c-e) Extracted neutral point of gate voltage (V_G), carrier density (n), and Fermi level (E_F) (Supplementary Fig. 22c-e).

Response to the Reviewer #3

This manuscript reports the CVD growth of twisted bilayer graphene with a range of interlayer twist angles. Twisted bilayer graphene has excited a lot of interest recently, and the development of bottom-up strategies for growing bilayer graphene (as well as other 2D materials) with controlled twist angles would be an important step forward in this field. Here, the authors report a first step, namely overcoming the preferred (equilibrium) stacking of the second layer during growth. However, clearly several challenges remain. For example, the twist angles obtained are random and cannot be selected at this point. And very small interlayer twists, where much of the exciting many-body physics is observed, are not achievable since the second layer nucleus would likely snap into registry with the first layer. Nevertheless, the study is quite carefully conducted and should be of interest to experts in the field, and can therefore be considered for publication in Nature Communications. However, the paper contains a number of weak points and issues that are mandatory to be addressed conclusively before acceptance may be considered. A list of the most urgent issues follows below:

Response:

We appreciate very much for the reviewer' positive comments on the quality of our work and explicit recommendation of publication. We will address the reviewer's comments point by point as follows.

Question 1:

Title, abstract, and text: The term "ex-situ nucleation" is unfortunate, since "ex-situ" implies outside of the growth system. This will certainly lead to misunderstandings, and could negatively impact the paper. I strongly urge the authors to find a better term for their approach, which should emphasize the main characteristics, namely that nucleation is forced by a gas pressure spike. Perhaps "far from equilibrium" could be used.

Response:

Thanks for the valuable suggestion. We also agree with the reviewer that the term “ex-situ” might cause the misunderstanding. Here, we want to express a concept that the nucleation site of the second layer graphene is different from that of the first layer, which is caused by a gas-flow perturbation. Thus, we have replaced the “ex-situ nucleation” by “hetero-site nucleation”. The hetero-site nucleation would allow second layer graphene nucleate at the site with different chemical environment, and drive the entire growth system out of equilibrium. All above the thermodynamics and kinetics factors would together contribute to the formation of interlayer twist, please also referring to the answers to Question 3 and Question 5.

Question 2:

Page 3, top: In the discussion of ways to overcome the equilibrium stacking registry, other approaches need to be included, even if they involve different 2D crystals other than graphene. This should include Ahn et al., Science 361, 782 (2018), where a sacrificial hBN layer on SiC was used to obtain 30 deg twisted bilayer graphene. And the discussion needs to include Sutter et al., Nature Commun. 10, 5528 (2019), where nucleation and transformation of an intermediate 2D crystal was used to obtain 30 deg twisted SnS2?

Response:

We are grateful for the reviewer’s valuable comments. Thanks to the reviewer’s suggestion to strengthen the conclusion of our manuscript, we carefully read the mentioned literatures, and revised our manuscript by adding corresponding reference as follow:

...Consequently, an approach for producing large-area, high-quality tBLG, with a full range of θ from 0° to 30° would significantly improve the ability to investigate its unique physical properties and applications. To achieve this, artificially-stacking methods have been developed, but suffers from unavoidable interlayer contamination which affects the interlayer coupling. Prior investigations indicated that azimuthally disordered multilayer graphene could be grown on C-terminated SiC.

Recently, 30°-tBLG was obtained by using a sacrificial hexagonal boron nitride (hBN) layer on Si-terminated SiC (Science **2018**, 361, 782). However, study and applications of the SiC-epitaxial tBLG are currently limited by the tedious transfer process and the relatively high price of SiC single crystal. In addition, the intermediate layer could also enable the formation of twist angle when growing the other two-dimensional (2D) materials (Nat. Commun. **2019**, 10, 5528).

Direct growth of tBLG on transition metal substrates such as Cu or Cu-Ni alloy via chemical vapor deposition (CVD) is currently considered one of the most promising methods, due to the high scalability and the excellent quality of CVD graphene. However, during the high-temperature CVD growth, the energetically favorable bilayer graphene structure is the non-twisted one, i.e., AB-stacked bilayer graphene (AB-BLG), whereas any rotation between the two layers would need to overcome a high energy barrier. Thus, compared to AB-BLG, the fraction of tBLG were limited to around 50% in previous reports. Thus, an efficient approach for growing tBLGs, especially in high fraction of twisted graphene and with a full range of twist angles, is still highly desirable.

Question 3:

The manuscript mentions “gas-flow turbulence” on several occasions, but there is no evidence that turbulence is indeed involved in the second-layer nucleation. Instead, it appears more likely just to be an effect of the rapid increase in the carbon supersaturation at higher precursor pressure, which forces the nucleation and does not allow it to take place in the otherwise preferred center of the flake. It appears that the gas spike drives the system far enough out of equilibrium that new nucleus does not adopt the preferred equilibrium stacking but grows with random orientation relative to the first graphene sheet. The authors need to carefully reconsider their explanation, especially a scenario that is based on non-equilibrium (kinetic) phenomena rather than “turbulence”. If they believe that the underlying cause is indeed gas turbulence (in the strict hydrodynamic sense), then evidence has to be provided supporting this conclusion.

Response:

We very much appreciate the reviewer’s comments and suggestions to strengthen our conclusion. We would like to answer this question as follows:

1) We apologies for the incurred misunderstanding by “gas-flow turbulence”. Indeed, the word “turbulence” is a concept in hydrodynamics. As mentioned by the reviewer, it is the sudden increase of carbon source supply (flow rate of CH₄) and partial pressure of hydrogen that is responsible for the new nucleation of second layer. The increase of partial pressure of hydrogen would enable the formation of hydrogen-terminated edges of the first layer graphene (*Chem. Sci.* **2014**, 5, 4639), which have two roles: 1) Impeding the attachment of active carbon species onto the edge of the first layer; 2) Decoupling the interaction of the first-layer graphene and Cu substrate, which promotes the diffusion of the carbon species beneath the first layer to fuel the growth of second layer (*J. Am. Chem. Soc.* **2014**, 136, 3040) (Figure R23). The sudden increase of carbon source supply produces an over-saturated supply of active carbon species near the edge of the first layer, which would contribute to the nucleation of second-layer graphene. The sudden increasing of H₂ and CH₄ should be a kind of gas-flow perturbation. Therefore, according to the reviewer’s suggestion, we replaced the word “**turbulence**” by “gas-flow **perturbation**” or “sudden increase of carbon source supply and partial pressure of hydrogen”.

Figure R23. The influence of sudden increasing partial pressure of hydrogen and carbon supply on the secondary nucleation beneath the first layer.

2) We understand and appreciate the reviewer’s kind suggestion very much on the explanation for the formation of twist bilayer graphene by “the gas spike drives the system far enough out of equilibrium that new nucleus does not adopt the preferred equilibrium stacking but grows with random orientation relative to the first graphene sheet”. We also agree with the reviewers that kinetic factors would drive the entire growth system out of equilibrium, in which the chance to form more stable AB-stacked bilayer graphene would be reduced, because the gas-flow perturbation is introduced within a short time. Besides kinetic factors, we also believe thermodynamic factors especially regarding the characters of Cu substrate play the main role in formation of the interlayer twist. Considering the

growth parameters of our experiment, the second layer graphene should nucleate and grow underneath the first layer, where the concentration of hydrogen is low. Therefore, the edge of the second layer graphene is passivated by metal atoms (Cu), resulting in a strong interaction between the second layer graphene and Cu substrate. Thus, the local chemical environment of the substrate plays crucial roles in determining the orientation of the second layer during the nucleation stage. The sudden increase of CH₄ and H₂ would allow the second-layer graphene nucleate at a different site from original one with a different local chemical environment, which would enable the formation of twist angle. We have included the possible explanation for the formation of tBLGs in Figure R24 and revised the corresponding discussion.

Figure R24. Possible causes of interlayer twist after introducing the gas-flow perturbation. Our main point is that the different local chemical environments of the two layers result in different thermodynamic preferred orientation. The kinetic phenomena induced by the gas spike can also induces the stacking disorder, but this is not the main reason.

3) **Evidences supporting our hypothesis on “different local chemical environments of nucleation sites are involved in the interlayer twist”.** Many aspects are related with the chemical environment of substrate, including the crystallographic orientation, steps, kinks, and particles. Experimentally, we added an example as shown in Figure R25. In brief, the surface roughness near the nucleation site of the second layer of graphene is higher than other regions (Figure R25), presumably owing to the presence of particles or kinks. Thus, the local environment (surface morphology) near the second-layer nucleation site should be different from that of the first layer graphene.

Detail experimental evidences and theoretical explanations on “gas-flow perturbation is indeed responsible for the second-layer nucleation” are provided in our response to **Question 5 of reviewer#1**.

Figure R25 (Supplementary Fig. 7). One example of substrate effect on the hetero-site nucleation of the second layer graphene. (a) SEM image of a tBLG. (b) Schematic diagram displaying the nucleation site of the second layer. (c-e) AFM height images zoomed in the region near the nucleation center. (f-h) AFM peak force error images corresponding to (c-e), which show clearer edges of the tBLG.

Question 4:

In the introduction and discussion, the authors need to clearly state the capabilities and limitations of the discussed method. While the CVD method can produce bilayer graphene with random twist angles, it is not able to reliably make bilayers with small twist angles (e.g., near the magic angle) and it is not (yet?) able to control the twist. This needs to be discussed especially carefully in the Discussion section, to inspire follow-up work addressing these limitations. Also, in the discussion, to

contrast with their results the authors should discuss and cite competing approaches aimed at bottom-up synthesis of interfaces with small twist angles, e.g., using Eshelby twist in nanowires (Nature 570, 354 (2019), and Nature 570, 358 (2019)).

Response:

We are grateful for the reviewer's valuable comments. According to the reviewer's suggestion, we revised the discussion of CVD method in the introduction and discussion sections, and cited corresponding literature accordingly to provide a comprehensive picture of methods for preparing the tBLGs.

At the end of the introduction section, sentence *"This new synthesis strategy by the intentional design of nucleation sites and would bring more inspiration for controlled growth of graphene and other 2D materials with precise control over interlayer twist in the future"* was added.

In the discussion part, we revised our manuscript as follow:

This work describes a hetero-site nucleation strategy for successfully growing the tBLG with the fraction of tBLG to BLG higher than 86%, and a wide range of twist angle from 0° to 30°. By employing the gas-flow perturbation after the nucleation of the first layer, the second layer nucleation could be initiated at a different location from that of the first layer, which was carefully investigated by isotope labeling in conjunction with micro-Raman spectroscopy. The sudden increase of carbon source supply would drive the system out of equilibrium, and allow the formation of the second-layer nuclei without the preferred equilibrium stacking order. In this work, since the second layer graphene is below the first layer, substrate plays crucial roles in determining the crystal orientation of graphene; the different local chemical environments of the nucleation sites of the two graphene layers are mainly responsible for the interlayer twist. The high quality of the as-grown tBLG was corroborated by clear Moiré patterns in the HR-TEM and carrier mobility exceeding $68,000 \text{ cm}^2 \text{ V}^{-1} \text{ s}^{-1}$ at room-temperature.

There are some issues still need to be addressed in the future works. Firstly, although by intentional design of nucleation sites, the hetero-site nucleation strategy enables the formation of small twist angles ($\sim 3^\circ$). The torque toward smaller twist angles increases rapidly near the zero-angle, which should be overcome to obtain the very small twist angle (such as magic angle). Our work brings more

*inspiration for controlled growth of graphene and other 2D materials with interlayer twist, and future works still need be done to grow tBLGs with certain twist angles, possibly by controlling the nucleation density, designing the growth substrate, utilizing the axial screw dislocation (Nature **2019**, 570, 358; Nature **2019**, 570, 354) or introducing intermediate layers (Nat. Commun. **2019**, 10, 5528; Science **2018**, 361, 782).*

Question 5:

Page 6: In the discussion of the second-layer growth, it should be clearly stated if the second layer grows above or beneath the first graphene layer. As far as I could see, this question is never addressed in the manuscript, while the Supplement indicates nucleation of the second layer beneath the first. This is a key point, since the nucleation of the second layer, forced by the increased supersaturation, appears to occur near the edge of the first and also without any fixed lattice registry with the first layer. This could give important hints for future approaches to control the twist angle, which the authors may want to mention in the Discussion. Related to this point, I would caution against using the terms “Stranski Krastanov” and “Volmer Weber” growth in the Supplement, since these have other meanings than those implied by the authors.

Response:

We are grateful for the reviewer’s valuable comments. We also agree with the location of second layer graphene is key point for potential readers. In our case, because the presence of high partial pressure of hydrogen would enable the formation of H-terminated edge of first-layer graphene, the carbon atom can diffuse under the first layer graphene (*J. Am. Chem. Soc.* **2014**, 136, 3040). Consequently, based on our experimental condition, we strongly believe the second layer graphene locate underneath the first layer. Therefore, we revised the schematic diagram of Figure 1a by adding a side view picture.

Revised figure 1a

Accordingly, we revised the manuscript by adding sentences at the beginning of the experimental section as follow:

The hetero-site nucleation strategy. During the high-temperature CVD process, high partial pressure of H_2 results in the H-terminated edges of graphene with reduced interaction with Cu substrate, which promotes the diffusion of the carbon species beneath the first layer to fuel the growth of the second layer graphene. In the conventional BLG growth experiment, both graphene layers would share the same nucleation site and grow simultaneously without introducing the gas-flow perturbation. In this case, the same surrounding microscopic environment, including Cu steps and particles, would results in the same crystalline orientation of the two layers, thus AB-BLG with no interlayer rotation is preferentially formed (Supplementary Fig. 1a). In contrast, after nucleation of the first graphene layer, subsequent nucleation of the second graphene layer is initiated by introducing a gas-flow perturbation, and therefore the nucleation of the second layer occurs at a distinct site, i.e., hetero-site nucleation (Fig. 1a). Therefore, the orientation of new layer would be determined by a different local environment, enabling the presence of interlayer twist and the formation of tBLGs. Note that the sudden carbon source supply enhancement was used as gas-flow perturbation, which is previously reported to be capable of initiating new nucleation (Supplementary Fig. 1b).

We revised the manuscript in Page 6, where we discuss the second-layer nucleation and growth:

Generally, when the concentration of active carbon species is enhanced, the nucleation rate and the growth rate would be increased (Adv. Mater. **2016**, 28, 4671; Nano Res. **2016**, 10, 355; Adv. Sci. **2018**, 5, 1700961). However, because the partial pressure of H_2 is greatly enhanced simultaneously, the graphene edge would be terminated by hydrogen, which hinders the attachment of the active

carbon species to the edge of the first layer (ACS Nano **2011**, 5, 6069; Nano Res. **2016**, 10, 355). According to the theoretical calculation, only carbon atoms can diffuse under the first layer to fuel the growth of the second layer graphene (J. Am. Chem. Soc. **2014**, 136, 3040; Nat. Nanotechnol. **2016**, 11, 426). Thus, if the nucleation of second layer occurs at the same site with that of first layer, several energy barriers must be overcome (Supplementary Fig. 6): 1) diffusion barrier-I corresponding to energy barrier needed to be overcome when carbon atoms diffuse from the bare Cu surface to graphene-covered region, 2) diffusion barrier II needed to be overcome when carbon atoms diffuse beneath the first layer graphene, and 3) nucleation barrier of the second layer graphene. The diffusion barrier-I is reported to be ~0.6 eV according to previous work (J. Am. Chem. Soc. **2014**, 136, 3040) so that it will impede the diffusion of carbon atom to the graphene-covered region, and increase the concentration of carbon atoms near the first layer graphene edge, which in turn promote the nucleation near the graphene edge. Near the edge of first layer, the high-energy sites (active sites) of the substrate in presence of steps, kinks, and particles would capture the C atoms and result in the nucleation of second layer, consistent with the reported literature (Nat. Commun. **2013**, 4, 2096). Therefore, when gas-flow perturbation is introduced, the second layer preferentially nucleates near the growth frontier of the first layer, rather than at the original nucleation center, and the local environments near the nucleation sites of the two graphene layers are therefore different, which is responsible for the formation of interlayer twist (Supplementary Fig. 7). In addition, the nucleation site of the second layer graphene can be controlled by the timing of the flow perturbation introduction, which agrees with the results in Fig. 2g.

Thanks for the reviewer's caution on using the terms "Stranski Krastanov" and "Volmer Weber" growth in the Supplementary Note 1. These words are indeed inappropriate here. We reconsidered the description of the two modes of growing bilayer or few-layer graphene, and revised the **Supplementary Note 1** accordingly:

*The growth of bilayer graphene or few-layer graphene (FLG) follow one of two modes: 1) **Mode-I**, the secondary-layer islands grow on the top layer of the first-layer graphene; 2) **Mode-II**, the secondary-layer islands grow beneath the first layer, in which the growth behavior is strongly influenced by the Cu substrate. Generally, if the carbon-source supply is extremely high, the growth of secondary-layer graphene would follow the first model, with irregular morphologies. If the carbon source supply is low or partial pressure of H₂ is relatively high, the graphene edge is usually terminated by hydrogen, and the decomposed carbon atoms can diffuse underneath the first layer*

graphene. In this case, considering the etching effect of hydrogen, the as-received graphene domains usually exhibit regular hexagonal shape.

Question 6:

Page 8: The authors briefly mention the formation of a quasicrystal at 30 degrees twist, but again without citing the Science paper by Ahn et al., which demonstrated this Stampfli tiling for the first time, as well as Sutter et al., which showed the quasicrystal for metal dichalcogenides (see Point (2), above). The prior work needs to be cited here, and the discussion of the quasicrystal tiling should be expanded slightly to better explain it to the reader.

Response:

We are grateful for the reviewer's valuable comments and apologies for the missing citations. According to the reviewer's suggestions, we added some discussions and cited corresponding literature accordingly, please referring as follow.

*The signature 12-fold rotational symmetry is also clearly visible in the as-grown 30° tBLG, where the Stampfli tiles, including equilateral triangles, squares and rhombuses can fill the entire space with different orientations (Supplementary Fig.19) (Science **2018**, 361, 782; Nat. Commun. **2019**, 10, 5528). The rotational order but lack of translational symmetry indicates that a high-quality quasi-crystalline system was formed.*

Question 7:

Methods: Replace “under 80 kV” and “under 2 kV” by “at 80 keV electron energy” etc.

Response:

We are grateful for the reviewer's valuable suggestions. we have revised our manuscript accordingly.

Question 8:

Figure 3 (d): I strongly suggest adding a second histogram, magnified to cover only the region below 5 degrees twist angle, with much finer binning. In this way, the authors can document clearly which twist angles are difficult to obtain (near 0 degrees), due to the tendency of the second layer to snap into registry with the first. Also, there is an apparent contradiction between the histograms in Fig. 3 (d) and Fig. 4 (c). In one case, a lot of 0 degree stacking is found, while the other figure claims almost no 0 degree stacked flakes. Please clarify!

Response:

We are grateful for the reviewer's valuable comments. For the twist angle below 2° , we also agree with the reviewer that “*due to the tendency of the second layer to snap into registry with the first*” and the presence of torque toward smaller twist angles, the tBLG with very small twist angle is difficult to grow. The figure 3d and figure 4c are based on OM image and SAED results, respectively. For OM-based method, we found difficult in distinguishing the AB-stacked bilayer graphene from tBLG with very small twist angle. In contrast, the error of determining the twist angle by SAED can be less than 0.5° , so that we can easily distinguish monolayer graphene, AB-BLG, and tBLG by SAED patterns (Figure R26). (Details on the error bar analysis please see the answers to the question 1, Reviewer#2).

Therefore, in our original manuscript, the AB-stacking counterpart is included in the column of the $0-2^\circ$ in Figure 3d (OM-based method). In contrast, we show the twist angle distribution of tBLG in the Figure 4c, and the number of AB-BLG is included in the pie chart, because we can distinguish AB-BLG from tBLGs with small twist angle (inset of Figure 4c).

Figure R26. SAED patterns and corresponding analysis of monolayer graphene (a), AB-BLG (b) and tBLG (c).

According to the kind suggestion of the reviewer, we added a column representing AB-BLG in figure 4c, and the bin width is set to **bin width is set to 1°**. In addition, to address the reviewer's concern about the distribution below 5°, we have added a second histogram (bin width is set to 0.5°) to supporting information (Supplementary Fig. 18 (Figure R27)). Considering the different inaccuracy error of OM-method and SAED-method, to avoid misunderstanding, we have put the statistical data obtained from OM characterization into supporting information (Supplementary Fig. 13 (Figure R28))

Figure R27(Figure 4c and Supplementary Fig. 18). Statistical results of twist angles. (a) Statistical results of stacking order (AB stacking or non-AB stacking) and distribution of twist angles based on SAED patterns of as-grown BLGs. **(b)** A second histogram which is magnified to cover only the region below 5° twist angle, with the bin width of 0.5°.

Figure R28 (Supplementary Fig. 13). Ratio of tBLG and AB-BLG and distribution of twist angles in as-grown tBLGs obtained by measuring the edges of hexagonal graphene domains using OM results. (a,b) Pie charts of stacking order (AB-BLG or tBLG) for graphene growth without hetero-site nucleation strategy (a) and the corresponding distribution of twist angles (b). (c,d) Pie charts of stacking order (AB-BLG or tBLG) for graphene growth with hetero-site nucleation strategy (a) and the corresponding distribution of twist angles (b).

Question 9:

The device fabrication still involves a significant amount of impurities and bubbling, as shown in Supplementary Fig. 14, which in a way runs counter to the idea of a clean bottom-up synthesis. The authors may want to include a brief discussion/outlook, stating how these issues could eventually be overcome in an all-synthesis approach to making encapsulated heterostructures.

Response:

We are grateful for the reviewer's valuable comments. We also agree with the reviewer's that the bubbling and impurities should be removed or avoided. The common way to reduce the contamination is

the high-temperature annealing. Furthermore, according to the previous reports, the contamination might come from the transfer step and high-temperature growth step (*Nat. Commun.* **2019**, 10, 1912.), thus, using the strategies for growing clean tBLGs or no polymer transfer would contribute to the removal of contamination. According to the reviewer's concern, we have included above discussion into our revised manuscript (Please see the Methods part).

In all, we appreciate having highly constructive remarks from all reviewers. With all their comments addressed in the text, we are looking forward to the publication of our manuscript in *Nature Communications*.

Sincerely yours,

Zhongfan Liu

List of changes

1. “*ex situ* nucleation” was replaced by “hetero-site nucleation” in the title and throughout the manuscript;
2. 4 coauthors were added in the author list, because of their contributions on the sample characterization.
3. The description of the second sentence of abstract was revised;
4. We revised the clarified corresponding expressions of “the portion of tBLGs in BLGs” across the entire manuscript, and the value 86% is replaced by 88%;
5. Sentences and new citations were added at the end of the first paragraph of Introduction;
6. The capabilities and limitations were stated at the end of the Introduction and Discussions;
7. The growth mode that the second layer graphene growth beneath the first layer was clearly stated at the beginning of the Results;
8. Theoretical analysis and experimental evidences were added in the revised manuscript when discussing the mechanism of hetero-site nucleation;
9. Description of quasi-crystalline and citations were added;
10. Description of ARPES experiments was added;
11. More comprehensive picture of methods for preparing the tBLGs were provided in the Discussions part;
12. A brief discussion on contamination was added in the Methods part;
13. A side view of schematic diagram was added in Fig. 1a;
14. A series of photographs of tBLGs were replaced in Fig. 1b;
15. Fig. 2 was revised by putting the statistical data of tBLGs obtained from OM-method to Supplementary Fig. 13;
16. Supplementary Note 1 was rewrite;

17. Rules and error analysis for measuring the twist angles by using OM-method were provided in Supplementary Note 5, Supplementary Table 1, and Supplementary Fig. 9, 10;
18. Error analysis for measuring the twist angles by using SAED data were provided in Supplementary Note 6, Supplementary Table 2, 3 and Supplementary Fig. 17;
19. A schematic diagram of energy barrier during the diffusion and nucleation of carbon atoms was added in Supplementary Fig. 6;
20. SEM and AFM data for an example of hetero-site nucleation was added in Supplementary Fig. 7;
21. A second histogram of twist angle of as-grown tBLGs was added in Supplementary Fig. 18;
22. Extracted doping level was provided in revised Supplementary Fig. 22;
23. Micro-ARPES data was added in Supplementary Fig. 23 to demonstrate the twist-angle-dependent electronic properties of tBLGs;
24. The number of figures were changed accordingly.

REVIEWER COMMENTS

Reviewer #1 (Remarks to the Author):

The author added more discussion of the growth mechanism in the revised manuscript. New APRES data and a discussion of error also added.

The author mentioned that their method cannot be applied on "magic-angle" twisted multilayer graphene, as small twist angle BLG is hard to form. Then the key result of the manuscript is just to generate more portion of random tBLG than Ref. 17 and 28. From the revised manuscript, the significance of this 30% more random tBLG to the field of twistrionics is still unshown.

The only interesting part is this "turbulence" before and "perturbation" in the revised manuscript. The CVD growth field is troubled with repeatability because lacking insightful understanding of growth mechanisms. As this manuscript will focus on the growth method, then the growth mechanisms are the foremost part of this manuscript. Still, the theory the author applied in the comment and the manuscript is not convictive enough. The relation between "perturbation" and the formation of more tBLG is still vague. At least in Ref. 17 and 28, they found both temperature, hydrogen partial pressure, methane concentration, and the substrates have a strong influence on the tBLG. Different from Ref. 17 and 28, the "perturbation" in this manuscript is a nonequilibrium process, and both parameters mentioned above will change dynamically. Also, when changing the growth parameter, the feedback time of the machine will not be zero, which will influence the "perturbation" process. The feedback time/behavior of the machine will vary due to different feedback settings and the brand. Further investigations/discussions on the following issues are suggested to make:

1. How the pressure, temperature of the sample surface will change during the whole "perturbation" process?
2. How the dynamic of the gas mixture/concentration in the "perturbation" process is in the tube?
3. What is the real gas flow/temperature feedback of the flowmeter/furnace during the "perturbation" process?
4. What parameter/parameters is/are crucial to controlling the seeding and growth of tBLG during the "perturbation" process?

The author demonstrated that using OM method to determine the twist angle of tBLG is not reliable in some cases. Then replacing all of them by SAED seems a better solution.

According to the considerations mentioned above, I do not think the novelty and quality of this manuscript reach the request of Nature Communications.

Reviewer #2 (Remarks to the Author):

Previously, my primary concern was that experts in CVD graphene growth might have felt that presented turbulence-induced triggered growth of the second bilayer was not new approach. i.e. that it is done routinely, and that this was just the first paper to document this growth-modification fully. I was aware of many CVD 'ad hoc' approaches that make twisted bilayer formation more likely, but to my knowledge that were all far more random, and gave far lower % twisted bilayer yields.

Fortunately, I find now that all three reviewers accept this turbulence triggered-growth method as a clear novel advance. It fair to say there is some disagreement on if this advance is 'sufficiently novel'. While this minority opinion is a fair viewpoint, I think it must be balanced against the overall high-quality of the revised analysis and the strong, rapid need for better growth methods to advance possible twistrionics applications. While it does not provide a selective pathway to enhance 'magic angle' growths in the <2% region, similar novel electronic many-body physics is also being discovered at the higher twist-angles as well. This work will help accelerate these discoveries.

The revisions made resulted in an improved manuscript that make the precise novelty of this growth approach more transparent. All language overstating-controllability for specific twist angles was been removed. The extensive revision clarifies (1) the error on angle determinations, (2) estimates on how the twisted bilayer growth modifies the effective Fermi-level, and (3) demonstrates clearly that preference for twisted-bilayer vs AB stacked growth does not lower the cumulative % bilayer yield. I now consider this carefully prepared, topical work now ready for publication in Nature Communications.

Reviewer #3 (Remarks to the Author):

The revised version of this manuscript addresses most of my original comments and questions, and has as a result been significantly improved. However, there are a few residual issues the authors need to address before acceptance of the paper can be recommended. Once these remaining points have been addressed in a satisfactory manner, the paper should be ready for publication.

1)The authors have now changed the terminology regarding the nucleation process of a twisted second layer, which is now more consistent with the evidence. However, a key question remains to be discussed better: What microscopic effect avoids equilibrium (AB) stacking, so that a large proportion of the bilayer flakes show interlayer twist? In the revised Discussion, the authors state "...since the second layer graphene locates underneath the first layer, substrate plays crucial roles in determining the crystal orientation of graphene; the different local chemical environments of the nucleation sites of the two graphene layers are mainly responsible for the formation of interlayer twist." This statement appears very superficial, and it neither provides an explanation for the suppression of equilibrium stacking, nor for the pretty broad distribution of stacking angles. The authors should try to address this point better.

2)Page 11: The authors added micro-ARPES measurements, which is positive. However the new results are only mentioned but not discussed in the manuscript. A brief discussion of the ARPES results should be added.

3)My other suggestions are mostly aimed at improving the language used in some parts of the manuscript:

- Line 155: "...growth frontier...", replace with either "growth front" or simply "edge".
- Line 186: "...This remarkable increase...", remove the word "remarkable". It is not necessary but hypes the result without any benefit to the paper.
- Line 259: "...axial crew dislocation...", this should say "...axial screw dislocation...".
- Line 290: "...performed in the SPECTROMICROSCOPY, Elettra, Italy...", should read "...performed on the Spectromicroscopy beamline, Elettra synchrotron (Italy)."

Response to referees

Response to the Reviewer #1

The author added more discussion of the growth mechanism in the revised manuscript. New APRES data and a discussion of error also added.

The author mentioned that their method cannot be applied on “magic-angle” twisted multilayer graphene, as small twist angle BLG is hard to form. Then the key result of the manuscript is just to generate more portion of random tBLG than Ref. 17 and 28. From the revised manuscript, the significance of this 30% more random tBLG to the field of twistrionics is still unshown.

The only interesting part is this “turbulence” before and “perturbation” in the revised manuscript. The CVD growth field is troubled with repeatability because lacking insightful understanding of growth mechanisms. As this manuscript will focus on the growth method, then the growth mechanisms are the foremost part of this manuscript. Still, the theory the author applied in the comment and the manuscript is not convictive enough. The relation between “perturbation” and the formation of more tBLG is still vague. At least in Ref. 17 and 28, they found both temperature, hydrogen partial pressure, methane concentration, and the substrates have a strong influence on the tBLG. Different from Ref. 17 and 28, the “perturbation” in this manuscript is a nonequilibrium process, and both parameters mentioned above will change dynamically. Also, when changing the growth parameter, the feedback time of the machine will not be zero, which will influence the “perturbation” process. The feedback time/behavior of the machine will vary due to different feedback settings and the brand. Further investigations/discussions on the following issues are suggested to make:

Response:

We sincerely thank the reviewer for the great effort in reviewing our manuscript.

We strongly believe that our method of “hetero-site nucleation” for growing twisted bilayer graphene (tBLG) is meaningful and instructive for potential readers of graphene community. Firstly, the concept of hetero-site nucleation, which is triggered by the perturbation and capable of breaking the thermodynamic equilibrium to induce the formation of twist angle, can be extended to the growth of other twisted two-dimensional materials. Furthermore, combined with the approaches for improving the domain size of graphene single crystal, our strategy can be used to grow twisted graphene single

crystals with hundreds micrometer-sized domain. The reason of tuning growth parameters including “temperature, hydrogen partial pressure, methane concentration, and the substrates” and detailed strategy and mechanism for growing tBLGs was not given in Ref. 17, and as for the Ref. 28, the main concern is only on the tBLGs with 30° twisted angle. In contrast, in our work, we have achieved the growth of tBLGs with a wide range of twist angle, and a detailed discussion on the growth mechanism has been given. In the following parts, we will provide new results about the growth of tBLGs with large domain size, and also the updated growth mechanism, as suggested by the reviewers.

In detail, based on our hetero-site nucleation method, we can control the nucleation density of tBLG by introducing the tiny amount of oxygen during the annealing process (centimeter-scale monolayer graphene single crystals were realized by introducing oxygen, which is reported by *Science* **2013**, 342, 720). Currently, the growth of 230- μm sized tBLG can be successfully achieved (Figure R1), which is clearly larger than the few-micrometers domain size in Ref.17. We also believe that relying on the hetero-site nucleation strategy, tBLGs with larger domain size will be synthesized in the future through the efforts in further controlling the nucleation density of tBLGs. The growth of millimeter-sized tBLGs single crystals would generate great interest for the readers who have research interest on the devices that requires millimeter-level uniformity of the twist angle.

Figure R1. OM images of as-synthesized tBLGs with large domain size.

Moreover, in our hetero-site nucleation strategy, surface states of the Cu substrate play crucial roles in the formation of interlayer twist, where the interaction between C atom in graphene and Cu would determine the crystalline orientation of graphene. In the nucleation stage of the second layer, Cu atoms of the Cu step edges and C atoms of graphene would form strong bond. The atomic arrangement of Cu step would influence the corresponding bond angle, and therefore determine the orientation of as-grown graphene nuclei. High-index facets of Cu, with rich and complex arrangement of atoms, such as steps, kinks, could supply different Cu steps with different atomic arrangement of Cu atoms. Therefore, the precise control of twist angles could be achieved by combining our hetero-site nucleation strategy and the use of high-index Cu single crystals as growth substrate, and successful preparation of decimeter-sized high-index single-crystal Cu substrates has been recently achieved (*Adv. Mater.* **2020**, 32, 2002034; *Nature* **2020**, 581, 406).

We appreciate the reviewer's interests on the mechanism of "perturbation" during the CVD growth. In this work, we found that the sudden increase of H₂ and CH₄ is capable to induce new nucleation of the second-layer graphene, near the edge of the first layer, i.e. the hetero-site nucleation. In our work, the temperature is kept the same during the entire growth process, and only the flow rates of H₂ and CH₄ are changed when introducing the perturbation. The influence of flow rates of H₂ and CH₄ on the formation of second-layer-free and second-layer-rich graphene domain is carefully investigated, which clearly shows the suitable flow rates of H₂ and CH₄ for introducing the nucleation of second layer (Figure 3 in the main text). Next, we will address the reviewer's concern on the details of the "perturbation" point by point.

Question 1:

How the pressure, temperature of the sample surface will change during the whole "perturbation" process?

Response:

Thanks for the valuable comment and suggestion. Temperature was not change manually when introducing the "perturbation", and the temperature changes caused by heating the introduced cold gas-flow is very limited (*Phys. Chem. Chem. Phys.* **2015**, 17, 22832). Thus, there should be no change

of temperature when we introduce “perturbation”. In the low-pressure CVD system, the flow velocity of H₂ and CH₄ is very high, and the pressure of the sample surface is nearly the same with the pressure of the whole chamber. By investigating the pressure of our CVD system when introducing the perturbation, we plot the pressure and the gas-flow rate of H₂ and CH₄ as functions of time (Figure R2). As shown in Figure R2, the pressure of the chamber would reach the stability within 10 seconds after the perturbation.

Accordingly, the related text in the Methods section was revised, and the time evolution of pressure, gas-flow rate of H₂ and CH₄ was included in the revised Supplementary Information (Supplementary Fig. 25).

Figure R2. The pressure, gas-flow as functions of time during the perturbation. The right panel shows more details on the time evolution of gas pressure.

Question 2:

How the dynamic of the gas mixture/concentration in the “perturbation” process is in the tube ?

Response:

We appreciate the reviewer’s comments. Indeed, in the hetero-site nucleation process, both the concentrations of active carbon species and partial pressure of hydrogen are important for inducing the new nucleation.

Before the perturbation process, in our typical growth conditions, the flow rate of H₂ and CH₄ are 400 sccm and 0.4 sccm, respectively, and the total pressure of the CVD system is about 350 Pa. As for the concentration of active carbon species, after the nucleation of the first-layer graphene, the consumption of active carbon species near the edge of the first layer would result in a concentration gradient from the edge of graphene to bare Cu (Figure R3a,b). Note that, in the first step, since the partial pressure of H₂ is relatively low, the graphene edge is usually metal-passivated (*J. Am. Chem. Soc.* **2014**, 136, 3040).

Then, the flow rates of H₂ and CH₄ were set as 1000 sccm and 1 sccm to initiate the nucleation of second layer, respectively. In low-pressure CVD (LPCVD) system, the flow velocities of H₂ and CH₄ are around tens of meters per seconds (G. Li, *et al. Phys. Chem. Chem. Phys.* **2015**, 17, 22832), and therefore the mass transport of H₂ and CH₄ from the inlet port to the sample takes only 10⁻² - 10⁻¹ s, clearly less than the time required for the stabilization of pressure after the introduction of perturbation (several seconds according to Figure R2). Therefore, the decomposition of the newly introduced methane would take place immediately after the introduction of perturbation (Figure R3c). With the increased partial pressure of the hydrogen, the edges of the first layer graphene are gradually terminated by hydrogen, which hampers the growth of the first layer and provides alternative diffusion pathway beneath the first layer for the decomposed active carbon. Because of the diffusion barrier for active carbon species to diffuse across the edge of the first layer to the region underneath the first layer (Supplementary Fig. 6, Diffusion barrier-I), there would be a peak of concentration of the active carbon species near the edge of the first layer, and the gradient of concentration is much sharper than that in the first step (Figure R3d).

It is worth noting that the nucleation rate is balanced by the concentration of active carbon species, and the etching of hydrogen. In region I and III (Figure R3), due to the improved etching effect of active hydrogen upon graphene at a high partial pressure of hydrogen, the nucleation rate is low (region I and III in Figure R3d) (*J. Phys. Chem. C* **2011**, 115, 17782). However, since the region II is covered by first-layer graphene, the nucleation barrier in region II is not strongly influenced by the high partial pressure of hydrogen (Figure R3d), and is lower than those in region I and III. In this regard, the secondary-layer nucleation is favorable to occur at region II near the edge of first layer.

After the hetero-site nucleation, since the growth rate of first layer graphene is higher than that of second layer, the distance between edges of first-layer graphene and second layer graphene would

become larger with the graphene growth. Therefore, the active carbon species will diffuse underneath the first-layer graphene with hydrogen-terminated edges to fuel the growth of second layer graphene. In this stage, the growth of second-layer result in enhanced consumption of active carbon species (Figure R3e). Therefore, the spatial distribution of concentration of active carbon species is shown in Figure R3f, where the peak of the concentration in Figure R3d would reduce or even disappear.

Figure R3. Elementary steps and the variation of concentrations of H and active carbon species during the perturbation.

Question 3:

What is the real gas flow/temperature feedback of the flowmeter/furnace during the “perturbation” process.

Response:

We very much appreciate the reviewer’s comments. Please referring to the reply to question 1, the temperature of our system is kept constant during the perturbation, and the time-evolution of the

gas-flow rate and the pressure is shown in Figure R4 (also see Figure R2). The one of typical perturbations we introduced in our experiment for growing tBLGs is taken as an example.

Figure R4. The time-evolution of pressure, gas-flow during the perturbation.

Question 4:

What parameter/parameters is/are crucial to controlling the seeding and growth of tBLG during the “perturbation” process?

Response:

Thanks for the valuable comment. The partial pressures of hydrogen and carbon source are crucial for the hetero-site nucleation, and we carefully investigate their influence on the formation of second-layer graphene, as shown in Figure 3 in the main text. As suggested by the reviewer, we have highlighted the function of partial pressures (flow rates) of hydrogen and carbon source in the formation of second layer graphene in the revised version.

A series of experiments were conducted to improve the growth controllability with respect to the three conditions. The ratio of MLG domains (second-layer-free domains) to all graphene domains (R_{MLG}) as functions of the H_2 and CH_4 (carbon-source supply) flow rates reveal the presence of two regions, as separated by green dash line (Fig. 3a and Supplementary Fig. 8a,b). One region (left bottom) consists of the corresponding parameters (H_2 and CH_4 flow rates) that are suitable for

growing second-layer-free graphene domains, which was chosen during the first nucleation step (step I). The other region (top right) is composed of the parameters for growing second-layer-rich graphene domains, which was used during the second layer growth step (step II).

In the hetero-site nucleation process, the increase of both H_2 and CH_4 flow rates are important for seeding and growth of the second layer graphene. The ratio of bilayer graphene domains to all graphene domains (R_{BLG}) as a function of the ratio of H_2 flow rate in step II to that in step I ($H_{2 \text{ step II}}/H_{2 \text{ step I}}$) is displayed in Fig. 3b. Clearly, a higher $H_{2 \text{ step II}}/H_{2 \text{ step I}}$ in the flow perturbation is beneficial for increasing R_{BLG} , highlighting the importance of increase the H_2 partial pressure in step II. High H_2 partial pressures enables H-termination of the graphene edge, which allows more active carbon species to diffuse beneath the first layer of graphene to fuel the growth of the second layer graphene³⁶. Meanwhile, at the same partial pressure of hydrogen, R_{BLG} would be higher when the flow rate of CH_4 in step II is higher than that in step I (Fig. 3b). However, an excess partial pressure of H_2 or CH_4 (over 1200 and 1.5 sccm for H_2 and CH_4 , respectively) would induce undesirable FLG formation, and therefore slightly reduce the R_{BLG} (Supplementary Fig. 8c, d). Therefore, the flow rate ratios of $H_{2 \text{ step II}}/H_{2 \text{ step I}}$ and $CH_{4 \text{ step II}}/CH_{4 \text{ step I}}$ must be carefully controlled in the hetero-site nucleation step (Fig. 3b).

Fig. 3 | Key parameters for growing tBLGs. a, R_{MLG} as functions of the gas flow rates of H_2 and CH_4 , respectively. The red and blue blocks denote the second-layer-free and second-layer-rich regions, respectively. The horizontal and vertical coordinates are logarithmic, while the color varies linearly with the ratio of second-layer-free domain to all graphene domains. **b,** R_{BLG} as functions of the flow

rate ratio of H_2 step II / step I. Black, red, and blue curves represent different CH_4 flow rate ratio (CH_4 step II / step I) of 1.0, 2.5, and 4.0, respectively.

Question 5:

The author demonstrated that using OM method to determine the twist angle of tBLG is not reliable in some cases. Then replacing all of them by SAED seems a better solution.

Response:

Thanks for the valuable comment. We understand the reviewer's concern and agree that it is not precise way to determine the twist angle by measuring the direction of graphene domain edges in OM image, in comparison with the obtained twist angles from SAED patterns in TEM characterization. However, using domain edge to estimate the twist angle is rapid, nondestructive and compatible with subsequent device fabrication. In the last round of revision, we updated corresponding description of the OM method for determining the twist angle and corresponding errors of OM-based method. The OM images of tBLG with various twist angles in Figure 1b were also updated and the statistical data derived by OM-based method have been moved to supporting information in the last round of revision for avoid misunderstanding. In our opinion, the OM results shown in Figure 1b would well demonstrate the capability of our method to grow tBLGs with a wide range of twist angle.

Response to the Reviewer #2

Previously, my primary concern was that experts in CVD graphene growth might have felt that presented turbulence-induced triggered growth of the second bilayer was not new approach. i.e. that it is done routinely, and that this was just the first paper to document this growth-modification fully. I was

aware of many CVD 'ad hoc' approaches that make twisted bilayer formation more likely, but to my knowledge that were all far more random, and gave far lower % twisted bilayer yields.

Fortunately, I find now that all three reviewers accept this turbulence triggered-growth method as a clear novel advance. It fair to say there is some disagreement on if this advance is 'sufficiently novel'. While this minority opinion is a fair viewpoint, I think it must be balanced against the overall high-quality of the revised analysis and the strong, rapid need for better growth methods to advance possible twistrionics applications. While it does not provide a selective pathway to enhance 'magic angle' growths in the <2% region, similar novel electronic many-body physics is also being discovered at the higher twist-angles as well. This work will help accelerate these discoveries.

The revisions made resulted in an improved manuscript that make the precise novelty of this growth approach more transparent. All language overstating-controllability for specific twist angles was been removed. The extensive revision clarifies (1) the error on angle determinations, (2) estimates on how the twisted bilayer growth modifies the effective Fermi-level, and (3) demonstrates clearly that preference for twisted-bilayer vs AB stacked growth does not lower the cumulative % bilayer yield. I now consider this carefully prepared, topical work now ready for publication in Nature Communications.

Response:

We appreciate very much for the positive comments from the reviewer on the quality of our work and the recommendation of publication.

Response to the Reviewer #3

The revised version of this manuscript addresses most of my original comments and questions, and has as a result been significantly improved. However, there are a few residual issues the authors need to address before acceptance of the paper can be recommended. Once these remaining points have been addressed in a satisfactory manner, the paper should be ready for publication.

Response:

We are happy to see that most concern of the reviewer have been addressed in the last round of revision, and we appreciate very much for the reviewer' efforts in reviewing our manuscript and the recommendation of publication. We will address the reviewer's comments point by point as follows.

Question 1:

The authors have now changed the terminology regarding the nucleation process of a twisted second layer, which is now more consistent with the evidence. However, a key question remains to be discussed better: What microscopic effect avoids equilibrium (AB) stacking, so that a large proportion of the bilayer flakes show interlayer twist? In the revised Discussion, the authors state "...since the second layer graphene locates underneath the first layer, substrate plays crucial roles in determining the crystal orientation of graphene; the different local chemical environments of the nucleation sites of the two graphene layers are mainly responsible for the formation of interlayer twist." This statement appears very superficial, and it neither provides an explanation for the suppression of equilibrium stacking, nor for the pretty broad distribution of stacking angles. The authors should try to address this point better.

Response:

Thanks for the valuable suggestion. The orientation of graphene is determined during the nucleation process, during which the interaction between graphene and the substrate plays crucial roles. Firstly, we'd like compare three kinds of interaction in terms of energy (Figure R5): 1) the interaction between graphene edge and Cu atoms ($G_{\text{edge-Cu}}$); 2) the interaction between the plane of graphene to Cu substrate ($G_{\text{plane-Cu}}$); 3) the interlayer interaction between graphene plane to graphene plane ($G_{\text{plane-Gplane}}$).

According to the reported literature (*J. Phys. Chem. Lett.* **2012**, 3, 2822), the energy of $G_{\text{edge-Cu}}$ is 1.43 eV/edge atom, and the energy of $G_{\text{plane-Cu}}$ is 30-35 meV/ C atom. In contrast, the interaction energies of $G_{\text{plane-Gplane}}$ are 43.69 meV/ C atom and 39.7 meV/ C atom for AB stacking and twisted configurations, respectively (*Nanoscale* **2013**, 5, 6736), which means that energy for creating a twist angle from AB-stacked configuration is only about 4 meV/ C atom. The total energy of $G_{\text{edge-Cu}}$ is proportional to the number of edge C atoms of graphene edge, N_E , while both the energies of $G_{\text{plane-Cu}}$

and $G_{\text{plane}}-G_{\text{plane}}$ are proportional to the total number of carbon atoms N in graphene plane ($N_E = \sqrt{6N}$, which is estimated by counting the number of edge atoms of hexagonal graphene domain); therefore, the interaction of $G_{\text{edge}}-\text{Cu}$ would be the main factor that determine the twist angle when the sizes of graphene nuclei are small.

To determine the dominant orientation of a graphene nucleus, the fluctuation of the formation energy, ΔE of tBLGs with different orientations is important. Taking C_{54} and C_{1014} (the size of C_{1014} is ~ 5.6 nm) as examples, ΔE ($G_{\text{edge}}-\text{Cu}$ of C_{54}) = 5.18 eV and ΔE ($G_{\text{edge}}-\text{Cu}$ of C_{1014}) ≈ 100 eV, respectively. However, the energy difference between AB stacking bilayer graphene and twisted bilayer graphene ($\Delta E_{\text{AB-twist}}$) is only about 0.216 eV and 4.056 eV for C_{54} and C_{1014} , respectively. Clearly, $\Delta E_{\text{AB-twist}}$ is lower than $\Delta E(G_{\text{edge}}-\text{Cu})$ for carbon clusters, indicating that the interaction between Cu edge atoms and graphene edge atom would be mainly responsible for determine the orientation of second layer graphene. Therefore, the interaction from the Cu substrate would suppress the equilibrium AB-stacking configuration to form interlayer twist.

Figure R5. Energy comparison among three pairs of interactions. (a-c) Schematic diagrams and corresponding energies of three pairs of interactions. (d-f) Energy fluctuation in terms of graphene edge-Cu interaction for C_{54} (d), graphene face-Cu interaction for each atom (e), and graphene interlayer interaction for each atom (f). (*J. Phys. Chem. Lett.* **2012**, 3, 2822, *Nanoscale* **2013**, 5, 6736)

There exist steps, kinks, dislocations and particles on the Cu surface, which would serve as nucleation sites for bilayer graphene. These the various atomic arrangement of the Cu step edge ensure the capability to growth bilayer graphene with a wide range of twist angle. For the step-attached

nucleation along $(01\bar{1})$ direction on Cu(111) surface, two orientations (0° and 30°) are energetically favorable. When the carbon atoms of zigzag (ZZ) edge attached to the Cu(111) step, the orientation of graphene is 0° with regard to the orientation substrates, while the orientation of graphene is 30° with its armchair (AC) edge attached to the Cu step (*J. Phys. Chem. Lett.* **2014**, 5, 3093). For high-index facets, the atomic arrangements of edge atoms turn more complex, which is key for forming a wide range of interlayer twist (Figure R6) (*Inorg. Chem. Front.* **2020**, 10.1039/D0QI00923G). Admittedly, the in-depth mechanism is still very unclear at the moment, we believe the kinks on high-index Cu surface with various orientation is important for suppressing the AB-stacked equilibrium.

Furthermore, the particles on the pre-melting Cu substrate would also enable the formation of second layer graphene with rich twist angles, because of Euclidean geometry of particles (*Science* **2020**, 370, 442-445). According to the reported literature (*ACS Nano* **2015**, 9, 1506), the particles on Cu surface is commonly observed, and would move on the Cu surface during the growth of graphene. Note that, the nucleation near the particles is also very common due to reduced nucleation barrier. Therefore, if the nucleation site of the secondary layer is near the particles, and at different site from that of the first layer, their surrounding environments would be totally different. Euclidean geometry of particles would induce the interlayer twist and lead to a broad distribution of twist angles.

We have included the discussion above into our revised manuscript and supplementary information according to the reviewer's concern.

Figure R6. Possible structures of the graphene nuclei on Cu surface with steps, kinks, and particles, which reflect their orientation. (a) ZZ edge attached and AC edge attached structures on Cu(111) with $(01\bar{1})$ step. (b) Various orientations of nuclei on high-index Cu(hkl) with kinks. Inset

shows the atomic structure of a typical kinked facet: Cu(532) (*Inorg. Chem. Front.* **2020**, 10.1039/D0QI00923G). (c) Determination of orientation of nuclei near the particles.

Figure R7. Time evolution of the particles during the high-temperature growth of graphene. The particles on the surface of Cu foil will move around during the growth of graphene. Data are adapted from the reported literature (*ACS Nano* **2015**, 9, 1506.)

Figure R8. Time evolution of the particles during the high-temperature growth of graphene (zoom in from Figure R7). (a,b) The particles pointed by the red arrow and in red circle will move driven by the expanding graphene edge. (c,d) The particles pointed by the blue arrow have the trend moving to the edge of the graphene island. The particle pointed by green arrow is disappearing (d). Data are adapted from the reported literature (*ACS Nano* **2015**, 9, 1506.)

Accordingly, the related text in the discussion part was revised as follow:

*The interaction between Cu substrate and the second-layer graphene is much stronger than the interlayer interaction of graphene during the nucleation stage, which would suppress the equilibrium AB-stacking configuration (*J. Phys. Chem. Lett.* **2012**, 3, 2822; *Nanoscale* **2013**, 5, 6736). The existence of steps, kinks, and particles on Cu surface serve as different chemical environments near the second-layer nucleation sites, which is mainly responsible for the formation of interlayer twist.*

We also added a Supplementary Note 8: Discussion on the formation of interlayer twist in our revised Supplementary Information:

Three kinds of interactions are important in determining the orientation of graphene: 1) the interaction between graphene edge and Cu atoms ($G_{edge-Cu}$); 2) the interaction between the plane of graphene to Cu substrate ($G_{plane-Cu}$); 3) the interlayer interaction between graphene plane to graphene plane ($G_{plane-G_{plane}}$). The schematic diagrams and corresponding energies of these interactions are shown in Supplementary Fig. 24a-c, which reveal that interaction of $G_{edge-Cu}$ is clearly stronger than that of $G_{plane-G_{plane}}$. To determine the dominant orientation of a graphene nucleus, the fluctuation of the formation energy, ΔE of tBLGs with different orientations is important. Taking C_{54} and C_{1014} (the size of C_{1014} is ~ 5.6 nm) as examples, ΔE ($G_{edge-Cu}$ of C_{54}) = 5.18 eV and ΔE ($G_{edge-Cu}$ of C_{1014}) ≈ 100 eV, respectively¹. However, the energy difference between AB stacking bilayer graphene and twisted bilayer graphene ($\Delta E_{AB-twist}$) is only about 0.216 eV and 4.056 eV for C_{54} and C_{1014} , respectively²¹. Clearly, $\Delta E_{AB-twist}$ is lower than $\Delta E(G_{edge-Cu})$ for carbon clusters, indicating that the interaction between Cu edge atoms and graphene edge atom would be mainly responsible for determine the orientation of second layer graphene. Therefore, the interaction from the Cu substrate would suppress the equilibrium AB-stacking configuration to form interlayer twist.

In another hand, there exist steps, kinks, dislocations and particles on the Cu surface, which could serve as nucleation sites for bilayer graphene. These the various atomic arrangement of the Cu step edge ensure the capability to growth bilayer graphene with a wide range of orientation. For the step-attached nucleation along (01 $\bar{1}$) direction on Cu(111) surface, two orientations (0° and 30°) are energetically favorable (Supplementary Fig. 24d)². For high-index facets, the atomic arrangements of edge atoms turn more complex²², which is key for forming a wide range of interlayer twist (Supplementary Fig. 24e). Admittedly, the in-depth mechanism is still very unclear at the moment, we believe the kinks on high-index Cu surface with various orientation is important for suppressing the AB-stacked equilibrium. The nucleation near the particles on pre-melting Cu substrate is also very common, because of the reduced nucleation barrier. These particles would also enable the formation of second layer graphene with rich twist angles, because of Euclidean geometry of particles²³. In addition, the particles on Cu surface is commonly observed, and would move on the Cu surface during the growth of graphene²⁴. Therefore, if the nucleation site of the secondary layer is near the particles, and at different site from that of the first layer, their surrounding environments would be totally different (Supplementary Fig. 24d-f).

Question 2:

Page 11: The authors added micro-ARPES measurements, which is positive. However the new results are only mentioned but not discussed in the manuscript. A brief discussion of the ARPES results should be added.

Response:

We are grateful for the reviewer's valuable comments. We added a brief discussion of the ARPES results in the main text, and corresponding detail discussion in Supplementary Note 7.

The brief discussion in revised main text is as follow:

We further investigated the electronic band structures of as-grown tBLGs ($\sim 3^\circ$, $\sim 6^\circ$, and $\sim 11^\circ$) by using the angle-resolved photoemission electron spectroscopy with submicrometer spatial resolution (micro-ARPES). The θ -dependent vHSs are clearly observed in the energy-momentum-dispersion diagrams along with the corresponding integrated energy distribution curves (EDS) (Supplementary Fig. 23).

Added Supplementary Note 7:

Supplementary Fig. 23 shows the micro-ARPES data collected from three obtained CVD-grown tBLG domains on Cu substrate. By overlapping the Brillouin zones of graphene layers with ARPES intensity maps, the twist angle θ of $\sim 3^\circ$, $\sim 6^\circ$, and $\sim 11^\circ$ ($\pm 1^\circ$) can be determined, respectively (Supplementary Fig. 23a-c). We also plotted the schematic diagrams of band structure for these tBLGs near the Dirac cones to show the twist-angle dependent band structure (Inset of Supplementary Fig. 23a-c). The vHSs arising from the interlayer coupling in band-crossing area are clearly observed in the energy-momentum-dispersion (the left panel of Supplementary Fig. 23d-f), where the top layer displays a relatively higher intensity in each figure. The corresponding integrated energy distribution curves (EDCs) are also plotted (right panel of Supplementary Fig. 23d-f) to quantitatively determine the position of vHSs. Considering the substrate doping effect on the tBLGs, the ΔE_{vHS} can be calculated by the formula $\Delta E_{\text{vHS}} = 2|E_{\text{D}} - E_{\text{vHS}}|$, where E_{D} is the energy of Dirac point. Clearly, as the θ increases, the ΔE_{vHS} increases as well, consistent with previous reports¹⁸. Our observation confirms that the

electronic structure of our tBLGs is twist-angle-dependent, which endows tBLGs the θ -dependent-enhanced optical absorption, θ -dependent-enhanced Raman G-band intensity and enhanced photocurrent generation at certain wavelengths^{7,19,20}.

Question 3:

My other suggestions are mostly aimed at improving the language used in some parts of the manuscript.

-Line 155: "...growth frontier...", replace with either "growth front" or simply "edge".

-Line 186: "...This remarkable increase...", remove the word "remarkable". It is not necessary but hypes the result without any benefit to the paper.

-Line 259: "...axial crew dislocation...", this should say "...axial screw dislocation..."

-Line 290: "...performed in the SPECTROMICROSCOPY, Elettra, Italy...", should read "...performed on the Spectromicroscopy beamline, Elettra synchrotron (Italy)."

Response:

We very much appreciate the reviewer's careful reading suggestions. We have revised these by 1) replacing "growth frontier" by "edge", 2) removing "remarkable", 3) replacing the word "crew" by "screw", and 4) rewriting the sentence "...performed on the Spectromicroscopy beamline..." as the reviewer suggested.

In all, we appreciate having highly constructive remarks from all reviewers. With all their comments addressed in the text, we are looking forward to the publication of our manuscript in *Nature Communications*.

Sincerely yours,

Zhongfan Liu

List of changes

1. Some language expressions were revised including 1) replacing “growth frontier” by “edge”, 2) removing “remarkable”, 3) replacing the word “crew” by “screw”, and 4) rewriting the sentence “...performed on the Spectromicroscopy beamline...” as the reviewer suggested.
2. Key parameters in the hetero-site nucleation process (perturbation) are stressed in the revised manuscript.
3. Figure 3b was revised, where the curves are more clearly marked than before.
4. A discussion of the ARPES results in the main text and Supplementary Note 7.
5. The microscopic effects on the formation of interlayer twist was added in the discussion part of revised manuscript and Supplementary Note 8 in the revised Supplementary Information.
6. A new Figure was added in the revised Supplementary Information (Supplementary Fig. 24).
7. Time evolution of gas-flow and pressure during the perturbation was added in the revised Methods section and revised Supplementary Information (Supplementary Fig. 25).

REVIEWER COMMENTS

Reviewer #1 (Remarks to the Author):

In Response to referees, the author tries to address my comments. However, it is still not enough convincing. In the manuscript, the edge nucleation is only attributed to the concentration of active carbon species and the edge's hydrogen termination. So, other possible influences from parameters, e.g., temperature, pressure, and feedback fluctuation, should be excluded. The author puts Phys. Chem. Chem. Phys. 2015, 17, 22832 to support there is no change of temperature when introducing "perturbation". However, in this paper, the simulation setups are different from what the author used in the experiment. In general, when the tube's diameter or the flow rates/pressure changed, the temperature distribution will be significantly different. So, it cannot be used to support the author's point. On the other hand, Figure R2 shows almost no overshoot in the flowmeters, which is good (or due to the slow sampling rate?). However, if there is no temperature change, why the pressure takes $> 20s$ to reach the balance while $< 2s$ for the flowmeters? Consider the growth mechanism is one of the core results in the manuscript. The author should do more carefully investigation of it.

Reviewer #3 (Remarks to the Author):

I have carefully reviewed the latest revision of this manuscript, and I am pleased to find that the authors have addressed all of my prior concerns in a satisfactory manner. I therefore recommend acceptance of the manuscript for publication in Nature Communications.

Response to referees

Response to the Reviewer #1

In Response to referees, the author tries to address my comments. However, it is still not enough convincing. In the manuscript, the edge nucleation is only attributed to the concentration of active carbon species and the edge's hydrogen termination. So, other possible influences from parameters, e.g., temperature, pressure, and feedback fluctuation, should be excluded.

The author puts Phys. Chem. Chem. Phys. 2015, 17, 22832 to support there is no change of temperature when introducing "perturbation". However, in this paper, the simulation setups are different from what the author used in the experiment. In general, when the tube's diameter or the flow rates/pressure changed, the temperature distribution will be significantly different. So, it cannot be used to support the author's point.

On the other hand, Figure R2 shows almost no overshoot in the flowmeters, which is good (or due to the slow sampling rate?). However, if there is no temperature change, why the pressure takes > 20s to reach the balance while < 2s for the flowmeters?

Consider the growth mechanism is one of the core results in the manuscript. The author should do more carefully investigation of it.

Response:

We sincerely thank the reviewer for the great effort in reviewing our manuscript.

We appreciate and understand the reviewer's concern on the mechanism of "perturbation" during the CVD growth and the contribution of temperature, pressure and feedback fluctuation to the edge nucleation. Indeed, our CVD growth process is performed under the condition of high temperature and high vacuum. Directly experimentally monitoring the variation of temperature, gas mixture, and pressure near the sample surface is still challenging. Therefore, suggested by the reviewer and to address the reviewer's concern, we performed computational fluid dynamics (CFD) simulations to understand the temperature and pressure change during the "perturbation". The simulation setup is according to the CVD system we used for experimentally growing tBLG (Figure R1a). The settings of inlet gas-flow is based on measurement results from the flowmeters we used (HORIBA METRON, S48

32/HMT) during the perturbation (Figure R1b). The temperature of the boundary of hot zone is set as constant 1020 °C, according to our measurement results (reading from the thermocouples near the wall of quartz tube we used) during the perturbation (Figure R1c). The outlet boundary condition is based on the mechanical pump in our experiment (ULVAC, GCD-136X). Note that the ability of mechanical pump to pump the gas is defined as a constant volume per unit time (135 L/min), regardless of the gas density, which is related to the volume or the rotate speed of the mechanical pump.

Figure R1. Model and boundary conditions of our CFD simulation. **a**, Geometry of the tube reactor. **b**, Gas flow settings in our simulation, which is based on the measurement results from the flowmeters we used (HORIBA METRON, S48 32/HMT) during the perturbation. For the convenience of presentation, here we define the moment starting the perturbation (t_p) as the zero point of time. **c**, Measured temperatures obtained from three thermocouples (three hot zones) near the wall of tube reactor we used.

Next, we will address the reviewer's comments point by point in the following.

Question 1:

The author puts Phys. Chem. Chem. Phys. 2015, 17, 22832 to support there is no change of temperature when introducing "perturbation". However, in this paper, the simulation setups are

different from what the author used in the experiment. In general, when the tube's diameter or the flow rates/pressure changed, the temperature distribution will be significantly different. So, it cannot be used to support the author's point.

Response:

Thanks for the valuable comment on the influence of *temperature* on the edge nucleation. As mentioned above, the temperature near (outside) the wall of the quartz tube was kept a constant temperature of 1020 °C during the gas-flow perturbation (Figure R1c). We performed CFD simulations based on our experimental conditions, and the incoming gas was set to be room temperature (300 K).

The spatial distribution of temperature inside the quartz tube at $t = t_p - 1$ s, $t = t_p + 1$ s, $t = t_p + 3$ s, and $t = t_p + 20$ s are shown in Figure R2a-d (t_p is time when the perturbation was introduced). The temperature near the sample surface (~ 1 mm from the sample surface) and the center of the tube chamber as a function of time are shown in Figure R2e. Clearly, it was found that the temperature near the sample surface kept the same during the perturbation. At the center of the tube, there indeed exists a small rise of temperature (0.3°C), due to the low temperature of compressed gas-flow during the perturbation, which, however, is so small that negligible influence on the CVD graphene growth was introduced.

To further prove the above conclusion and exclude the influence of 0.3°C fluctuation, we conducted an experiment that we introduced 1°C fluctuation manually as “perturbation” during the graphene growth. After the manually introduced temperature fluctuation, additional 10 min-growth was proceeded for the “growth of second layer”. All the other conditions were kept the same. It was found that there is no nucleation of the second layer graphene during the CVD growth (Figure R3).

Figure R2. Temperature analysis during the perturbation. a-d, Spatial distribution of temperature in the tube reactor at $t = t_p - 1$ s (a), $t = t_p + 1$ s (b), $t = t_p + 3$ s (c), and $t = t_p + 20$ s (d), respectively. e, Temperature near the sample surface ($X = 0$, $Y = -3.4$ cm, about 1 mm from the sample surface) and temperature at the tube center ($X = 0$, $Y = 0$) as a function of time during the perturbation. t_p : the time when the perturbation was introduced.

Figure R3. Experimental results when manually introducing a fluctuation of 1°C. SEM images of graphene grown under a 1°C fluctuation, which shows no bilayer regions.

Question 2:

On the other hand, Figure R2 shows almost no overshoot in the flowmeters, which is good (or due to the slow sampling rate?). However, if there is no temperature change, why the pressure takes > 20s to reach the balance while < 2s for the flowmeters?

Response:

Thanks for the valuable comment. The flowmeters (mass flow controller, MFC) we used in our experiments are purchased from HORIBA METRON (S48 32/HMT). The flow rate measuring ranges of H₂ and CH₄ are 2000 and 10 standard cubic centimeter per minute (sccm), respectively. According to the manual of MFC (HORIBA METRON, S48 32/HMT), the response time of this series products is less than 2 s, and there is no overshoot.

Then, we'd like to explain why it takes ~20 s for the **pressure** to reach the new balance.

A stable pressure of our CVD system results from the balance of incoming and outgoing gases, which are controlled by the MFCs and mechanical pump, respectively. During the perturbation (sudden increase of gas-flow rate), the increase of the gas in the chamber would cause the increase of pressure.

For the gas flowing in, the flow rates of incoming gases rise to the target value within 2 seconds and then would be constant. Note that the MFC controls the flow rate according to the mass of flowing gas, i.e. it is the mass of the incoming gases that is set.

For the gas flowing out, however, the mass of the gases pumping out per unit time changes in a positive correlation with the chamber pressure. Usually, the ability of mechanical pump to pump the gas is defined as a constant volume per unit time, regardless of the gas density, which is related to the volume or the rotate speed of the mechanical pump. Therefore, the density or the pressure of remaining gas in the chamber determines the mass of the gas to be pumped per unit time. In this regard, during the perturbation, the increased pressure of the chamber result in a faster flowing out of the gas in the chamber, and a slower accumulation of the incoming gas in the chamber to reach the balance, and thus ~20 s is required to attain the stability of pressure during the perturbation. Finally, the pressure reaches a new steady value. To sum up, it is the feedback of “chamber pressure — pumping rate (mass flow rate) — chamber pressure” that results in the time delay.

CFD simulation results also prove our points. Figure R4a shows the pressure contour in the furnace before the gas perturbation ($t = t_p - 1$ s), in which pressure is uniform in the tube. After the

sudden increase of gas flow rate ($t = t_p + 1$ s, $t = t_p + 3$ s, and $t = t_p + 20$ s), we can observe an increase of pressure, which, however, still exhibited a uniform distribution (Figure R4b-d).

We extracted the value of pressure at the center of tube reactor ($X = 0$, $Y = 0$) and the pressure near the sample surface (1 mm), respectively, and plotted the pressure as a function of time when introducing the perturbation (Figure R4e). The pressure rises rapidly at first, but its increasing speed (slop) gradually decreases, which results in the formation of a new balance between the incoming and outgoing gases. Note that the simulation results are consistent with the experimentally measured results, and the real-time pressure would reach 80% of the target value within 5 seconds.

Figure R4. Pressure analysis during the perturbation. a-d, Spatial distribution of pressure in the tube reactor at $t = t_p - 1$ s (a), $t = t_p + 1$ s (b), $t = t_p + 3$ s (c), $t = t_p + 20$ s (d), respectively. e, Pressure as a function of time during the perturbation. t_p : the time when the perturbation was introduced.

Question 3:

Consider the growth mechanism is one of the core results in the manuscript. The author should do more carefully investigation of it.

Response:

Thanks for the valuable comment and suggestions. Based on the CFD simulations, we can also analysis the gas-flow dynamics on the gas mixture/concentration, and gas-flow velocity during the perturbation.

Gas mixture/concentration. As mentioned in the last round of peer review, the concentration of H₂, CH₄, active hydrogen, and active carbon species play crucial roles in the nucleation of the second-layer graphene during the perturbation. Utilizing the CFD simulation, we can obtain a full picture of gas mixture evolution in macro-scale during the perturbation.

Figure R5 show the simulation results of H₂ and active hydrogen (H), respectively. From Figure R5a, we can find that before the perturbation, the molar concentration of H₂ in the hot zone is lower than that outside the hot zone, which is consistent with the ideal gas law (the amount of substance $n = pV/RT$, where P , V , and T are the pressure, volume and temperature). Figure R5b shows the spatial distribution of the molar concentration of H near the sample surface (marked with the white dashed box in Figure R5a). It is found that the concentration of H near the sample surface is higher than that in the tube center, and increases downstream along the tube, which can be ascribed to the slow flow velocity in the boundary layer and the continuous decomposition of H₂ near the sample surface. After the rise of flow rate and pressure during the perturbation, the molar concentration of H₂ and H increases accordingly (Figure R5c and Figure R5d), and reach a balance after around 20s, consistent with the time evolution of pressure. Note that the real-time concentration would reach 80% of the target value within 5 seconds. Similarly, we also plot the molar concentration distribution of CH₄ and active carbon species (C_xH_y, x=1,2, y=0, 1, 2, 3, 4, 5) in Figure R6.

Figure R5. Concentration of H₂ and active hydrogen (H) during the perturbation. **a**, Spatial distribution of molar concentration of H₂ in the tube at $t_p - 1$ s, $t_p + 3$ s, $t_p + 6$ s, and $t_p + 20$ s, respectively. **b**, Spatial distribution of molar concentration of H near the sample surface (marked in panel **a**). **c,d**, Molar concentration of H₂ (**c**) and H (**d**) as functions of time. Note that the real-time concentration of H₂ and H would reach 80% of the target value within 5 seconds. t_p : the time when the perturbation was introduced.

Figure R6. Concentration of CH_4 and active carbon species (C_xH_y) during the perturbation. a, Spatial distribution of molar concentration of H_2 in the tube at $t_p - 1$ s, $t_p + 3$ s, $t_p + 6$ s, and $t_p + 20$ s, respectively. **b,** Spatial distribution of molar concentration of C_xH_y near the sample surface (marked in panel a). **c,d,** Molar concentration of CH_4 (c) and C_xH_y (d) as functions of time. Note that the real-time concentration of CH_4 and C_xH_y would reach 80% of the target value within 5 seconds. t_p : the time when the perturbation was introduced.

Gas-flow velocity. Indeed, the sudden increase of gas flow rate will cause the fluctuation of gas-flow velocity. Figure R7a shows the simulation results of velocity distribution in the tube reactor, where the region near the sample surface (marked with white dashed box) is enlarged (Figure R7b). We extracted the time-evolution of the gas-flow velocity at the center of tube reactor ($X = 0$, $Y = 0$) and the pressure near the sample surface (1 mm away from the sample surface) (Figure R7c). The gas-flow velocity increases rapidly at the beginning, and return to the original value within 5 s, and the

decreasing rate is smaller than the increasing rate at the beginning. Note that the velocity of the gas flow in the boundary layer is much lower than that in the tube center. Therefore, the fluctuation on the velocity near the sample surface would have little influence on the nucleation of the second layer graphene.

Figure R7. Gas-flow velocity analysis during the perturbation. a, Spatial distribution of velocity in the tube reactor at $t_p - 1$ s, $t_p + 1$ s, $t_p + 3$ s, and $t_p + 20$ s, respectively. **b,** Spatial distribution of velocity near the sample surface (marked in panel a). **c,** Gas-flow velocity as a function of time during the perturbation. t_p : the time when the perturbation was introduced.

Revisions in the manuscript and Supplementary Information

According to our simulation results, the related text in the *Mechanism of the hetero-site nucleation* was revised as follow:

Computational fluid dynamics (CFD) simulations were conducted to investigate the dynamics of the gas-flow during the perturbation. After the sudden increase of flow-rate of H₂ and CH₄, the gas-flow velocity increases rapidly at first, and return to the original value within 5 s (Supplementary Fig. 6a,b). Note that, the gas-flow velocity in the boundary layer is slow; therefore, the influence of the fluctuation of the gas-flow velocity near the sample surface on the nucleation of the second layer graphene is really limited. A uniform distribution of pressure across the entire the tube reactor is observed, and the uniformity retains after the increasing of the pressure caused by the perturbation. Because of the feedback of chamber pressure — pumping rate (mass flow rate) — chamber pressure, the increasing rate of the pressure would decrease gradually with the time: the real-time pressure reaches 80% of the target value within 5 s and finally reaches a new steady value in about 20 s (Supplementary Fig. 6e,f). The concentration of H₂, CH₄, active hydrogen, and active carbon species also increase accordingly (Supplementary Fig. 7).

Generally, when the concentration of active carbon species is enhanced, the nucleation rate and the growth rate would be increased^{30,31,35}. However, because the partial pressure of H₂ is greatly enhanced simultaneously, the graphene edges would be terminated by hydrogen.....

The method section is revised by adding the CFD simulation.

CFD simulation. The CFD model here is based on laminar compressible flow with surface chemical reaction. Settings on gas mixture, chemical reactions are based on the previous reported literature⁵³. CFD code ANSYS FLUENT is used here, and the computational mesh arrangement of over 200,000 cells is developed with fine mesh near the sample surface to guarantee the convergence. The geometry and boundary conditions of gas inlet and temperature are set as shown in Supplementary Fig. 27. The outlet boundary condition is based on the mechanical pump in our experiment (ULVAC, GCD-136X), which is set as 135 L/min.

Supplementary Fig. 6 and Supplementary Fig. 7 on the CFD simulation results in our revised Supplementary Information are as follow:

Supplementary Fig. 6: Computational fluid dynamics (CFD) simulation results on the dynamics of gas-flow velocity, temperature and pressure during the gas-flow perturbation. a, Spatial distribution of velocity in the tube reactor at $t = t_p - 1$ s, $t = t_p + 1$ s, and $t = t_p + 20$ s, respectively. **b,** The gas-flow velocity at the center of the tube and the velocity near the sample surface (~ 1 mm) as a function of time during the perturbation. **c,** Spatial distribution of temperature in the tube reactor at $t =$

$t_p - 1$ s, $t = t_p + 1$ s, and $t = t_p + 20$ s, respectively. **d**, Temperature as a function of time during the perturbation. Only a small rise of temperature (0.3°C) at the center of the tube and almost no temperature fluctuation near the sample surface (~ 1 mm from the sample surface) are observed, which would have negligible influence on the CVD graphene growth. **e**, Spatial distribution of pressure in the tube reactor at $t = t_p - 1$ s, $t = t_p + 1$ s, and $t = t_p + 20$ s, respectively. **d**, Simulated results of pressure (at the center of the tube and near the sample surface) and experimentally measured pressure as a function of time during the perturbation. Note that the real-time pressure would reach 80% of the target value within 5 seconds after the perturbation. t_p : the time when the perturbation was introduced.

Supplementary Fig. 7: CFD simulation results on the dynamics of gas mixture/concentration. a,b Molar concentration of H_2 (a) and active hydrogen H (d) as functions of time. c,d, Molar concentration of CH_4 (c) and active carbon species (C_xH_y) (d) as functions of time. t_p : the time when the perturbation was introduced.

Supplementary Fig. 27: Model and boundary conditions of our CFD simulation. **a**, Geometry of the tube reactor. **b**, Gas flow settings in our simulation, which is based on the measurement results from the flowmeters we used (HORIBA METRON, S48 32/HMT) during the perturbation. **c**, Measured temperatures obtained from three thermocouples (three hot zones) near the wall of tube reactor we used during the gas-flow perturbation. t_p : the time when the perturbation was introduced.

Response to the Reviewer #3

I have carefully reviewed the latest revision of this manuscript, and I am pleased to find that the authors have addressed all of my prior concerns in a satisfactory manner. I therefore recommend acceptance of the manuscript for publication in Nature Communications.

Response:

We thank the reviewer very much for the positive comments on the quality of our revision and the recommendation of publication.

In all, we appreciate having highly constructive remarks from all reviewers. With all their comments addressed in the text, we are looking forward to the publication of our manuscript in *Nature Communications*.

Sincerely yours,

Zhongfan Liu

List of changes

1. Description and analysis on computational fluid dynamics (CFD) simulations were added in the section of *Mechanism of the hetero-site nucleation*, and related methods were added in the *Methods* section.
2. CFD simulation results and related boundary conditions were added in Supplementary Information (Supplementary Fig. 6, Supplementary Fig. 7 and Supplementary Fig. 27).
3. Two coauthors were added in the author list, because of their contributions on the CFD simulations.
4. A sentence “A sudden increase of H₂ and CH₄ ...” was added in the first paragraph, which makes the perturbation more explicit when describing the strategy of hetero-site nucleation.
5. Figure numbers of Supplementary Information were changed accordingly.
6. Two citations were added.
7. Acknowledgement and Author contribution sections were revised by adding funding numbers and author contributions.

REVIEWERS' COMMENTS

Reviewer #1 (Remarks to the Author):

The author added computational fluid dynamics simulation to support their growth mechanism. The integrity of manuscript is improved, and I think it now meets the quality request of Nature Communications.

A few suggestions:

1. Setting the pumping rate according to the pump performance curve will be better.
2. In Fig. S6e, it will be better to have different color bars for each distribution. Now it is too large and wipe out the detail of pressure distribution.
3. I also suggest the author upload the COF computational file as source data file when it published.

Response to referees

Response to the Reviewer #1

The author added computational fluid dynamics simulation to support their growth mechanism. The integrity of manuscript is improved, and I think it now meets the quality request of Nature Communications.

A few suggestions:

Response:

We thank the reviewer very much for the positive comments on the quality of our revision and the recommendation of publication. We have revised our manuscript according to the reviewer's suggestions, and uploaded the computational files as source data file.

The point-to-point responses are as follow:

Question 1:

Setting the pumping rate according to the pump performance curve will be better.

Response:

Thanks for the valuable comment. We conducted the CFD simulation again. The outlet boundary condition is based on the pump performance curve (Figure R1), which is obtain from the specification of our mechanical pump (ULVAC, GCD-136X). And the related data in Supplementary Information (Figure S6 and S7) have been revised.

Figure R1. Pumping speed as a function of pressure of the chamber, which is obtained from the specification of our mechanical pump (ULVAC, GCD-136X). Pressure range from 200 Pa to 1100 Pa is our focus.

Question 2:

In Fig. S6e, it will be better to have different color bars for each distribution. Now it is too large and wipe out the detail of pressure distribution.

Response:

Thanks for the valuable comment. As we conducted the CFD simulation again based on the new outlet boundary condition. The data in Figure S6 are all revised as follow. According to the reviewer’s suggestion, the color bars are set different for the contours of pressure distribution. Note that, the pressure distribution in the quartz tube is very uniform (<1 Pa), and the pressures near the inlet port (diameter = 0.6 cm) is 30~50 Pa higher than those in the quartz tube. Therefore, we enlarge this region with new color bars for each time.

Figure R2. (Supplementary Fig. 6)

We also revised Supplementary Fig. 7 as follow:

Figure R3. (Supplementary Fig. 7)

Question 3:

I also suggest the author upload the COF computational file as source data file when it published.

Response:

Thanks for the valuable comment and suggestions. We have uploaded our CFD source files.

In all, we appreciate having highly constructive remarks from all reviewers. With all their comments addressed in the text, we are looking forward to the publication of our manuscript in *Nature Communications*.

Sincerely yours,

Zhongfan Liu

List of changes

1. CFD simulation results have been revised, and related boundary conditions (pumping rate is now according to the pump performance curve) were revised in *Methods* section and Supplementary Information (Supplementary Fig. 6, Supplementary Fig. 7 and Supplementary Fig. 27).
2. Considering the contributions on CFD simulations and further explanation on the simulations, Prof. Shenghong Huang is listed as a corresponding author.